# Tau filaments with the Alzheimer fold in human *MAPT* mutants V337M and R406W

Chao Qi[1], Sofia Lövestam[1], Alexey G. Murzin[1], Sew Peak-Chew [1],
Catarina Franco[1], Marika Bogdani[2,3], Caitlin Latimer[2,3], Jill R. Murrell[4,5],
Patrick W. Cullinane[6,7], Zane Jaunmuktane[6,7], Thomas D. Bird[2,3],
Bernardino Ghetti [4], Sjors H. W. Scheres [1,8] ✉ & Michel Goedert [1,8] ✉

Frontotemporal dementia (FTD) and Alzheimer's disease (AD) are the most common forms of early-onset dementia. Unlike AD, FTD begins with behavioral changes before the development of cognitive impairment. Dominantly inherited mutations in *MAPT*, the microtubule-associated protein tau gene, give rise to cases of FTD and parkinsonism linked to chromosome 17. These individuals develop abundant filamentous tau inclusions in brain cells in the absence of β-amyloid deposits. Here, we used cryo-electron microscopy to determine the structures of tau filaments from the brains of human *MAPT* mutants V337M and R406W. Both amino acid substitutions gave rise to tau filaments with the Alzheimer fold, which consisted of paired helical filaments in all V337M and R406W cases and of straight filaments in two V337M cases. We also identified another assembly of the Alzheimer fold into triple tau filaments in a V337M case. Filaments assembled from recombinant tau (297–391) with substitution V337M had the Alzheimer fold and showed an increased rate of assembly.

In the adult human brain, six tau isoforms are expressed from a single gene by alternative mRNA splicing[1]. They differ by the presence or absence of one or two inserts in the N-terminal half and an insert in the C-terminal half. The latter encodes a repeat of 31 amino acids, giving rise to three isoforms with four repeats (4R). The other three isoforms have three repeats (3R). Together with adjoining sequences, these repeats constitute the microtubule-binding domains of tau[2]. Some of the repeats also form the cores of assembled tau in neurodegenerative diseases, suggesting that the physiological function of microtubule binding and the pathological self-assembly are mutually exclusive. Most *MAPT* mutations are in exons 9–12, which encode R1–R4, with some mutations being present in exon 13, which encodes the sequence from the end of R4 to the C terminus of tau. Only two of the 65 known *MAPT* mutations impact residues near the N terminus of tau[3]. Mutations in

*MAPT* lead to the formation of filamentous inclusions that are made of either 3R, 4R or 3R + 4R tau[4]. Mutations that cause the relative overproduction of wild-type 3R or 4R tau result in the deposition of 3R tau with the Pick fold[5] or 4R tau with the argyrophilic grain disease (AGD) fold[6]. In cases of sporadic and familial tauopathies, filaments of transmembrane protein 106B (TMEM106B) also form in an age-related manner[7–9].

Structures of 3R + 4R tau-containing filaments from cases with *MAPT* mutations have not been reported. In sporadic diseases, filaments made of 3R + 4R tau have the Alzheimer[10] or the chronic traumatic encephalopathy (CTE)[11] fold. The Alzheimer tau fold is also found in familial British and Danish dementias, cases of prion protein amyloidoses and primary age-related tauopathy[6,12]. The CTE tau fold is also typical of subacute sclerosing panencephalitis, amyotrophic lateral sclerosis–parkinsonism dementia complex and vacuolar tauopathy[13–15].

[1]MRC Laboratory of Molecular Biology, Cambridge, UK. [2]Departments of Neurology and Pathology, University of Washington, Seattle, WA, USA. [3]Veterans Administration Puget Sound Health Care System, Seattle, WA, USA. [4]Department of Pathology and Laboratory Medicine, Indiana University School of Medicine, Indianapolis, IN, USA. [5]Department of Pathology and Laboratory Medicine, Children's Hospital of the University of Pennsylvania, Philadelphia, PA, USA. [6]Department of Clinical and Movement Neurosciences, Queen Square Institute of Neurology, University College, London, UK. [7]Queen Square Brain Bank for Neurological Disorders, Institute of Neurology, University College, London, UK. [8]These authors contributed equally: Sjors H. W. Scheres, Michel Goedert. ✉e-mail: scheres@mrc-lmb.cam.ac.uk; mg@mrc-lmb.cam.ac.uk

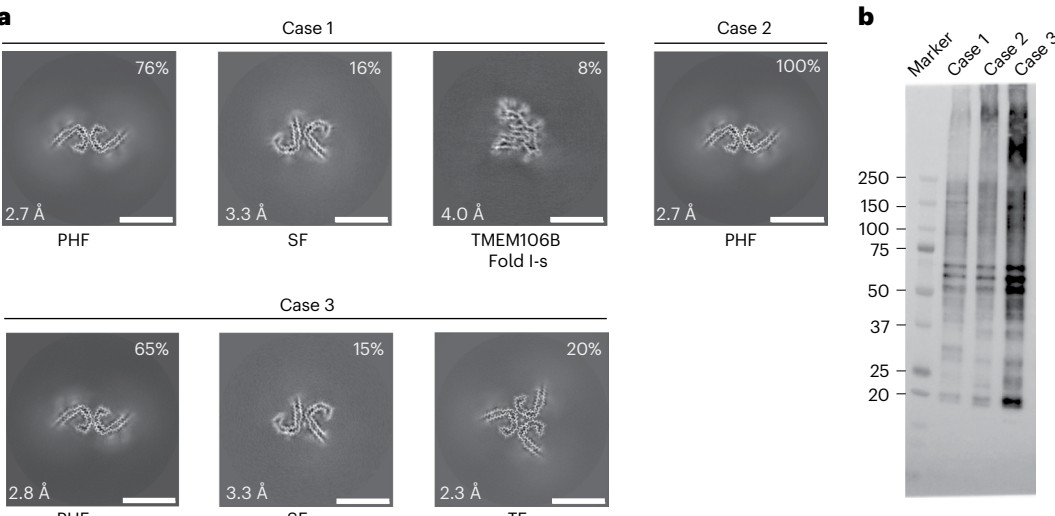

**Fig. 1 | Mutation encoding V337M in *MAPT*: cryo-EM cross-sections of tau filaments and immunolabelling. a**, Cross-sections through the cryo-EM reconstructions, perpendicular to the helical axis and with a projected thickness of approximately one rung, are shown for the frontal cortex from cases 1–3. Resolutions (in Å) and percentages of filament types are indicated at the bottom left and top right, respectively. Scale bar, 10 nm. **b**, Immunoblotting of sarkosyl-insoluble tau from the frontal cortex of cases 1–3 with substitution V337M. Phosphorylation-independent anti-tau antibody BR134 was used.

Recombinant tau (297–391) forms filaments with either fold, depending on the in vitro assembly conditions[16].

Dominantly inherited mutations encoding V337M (refs. 17–21) in exon 12 and R406W (refs. 22–28) in exon 13 of *MAPT* give rise to frontotemporal dementia (FTD) with inclusions that are also made of all six tau isoforms[18,24]. Amino acid substitution V337M, which is located inside the ordered cores of tau filaments[4], causes behavioral-variant FTD and cognitive impairment in the fifth or sixth decade[17,20,29]; it has been reported that tau inclusions are abundant in the cerebral cortex but not in the hippocampus[17]. Amino acid substitution R406W, which is located outside the ordered cores of tau filaments, is associated with an Alzheimer's disease (AD)-like amnestic phenotype that is characterized by initial memory impairment[30,31]; abundant tau inclusions are present in both the cerebral cortex and the hippocampus[22]. Here, we show that tau filaments from the brains of individuals with mutations encoding V337M and R406W in *MAPT* adopt the Alzheimer fold.

## Results

### Structures of tau filaments from three cases of Seattle family A with *MAPT* mutation encoding V337M

We used cryo-electron microscopy (cryo-EM) to determine the atomic structures of tau filaments from the frontal cortex of three previously described individuals of Seattle family A with a mutation encoding V337M in *MAPT* (Figs. 1 and 2 and Table 1)[17,20]. According to immunohistochemistry, abundant tau inclusions were present in the frontal cortex (Extended Data Fig. 1a)[17,18]. Unlike previous reports[17], we also detected hippocampal tau inclusions (Extended Data Fig. 1b).

Using cryo-EM, we observed the presence of the Alzheimer tau fold in all three cases (Figs. 1 and 2)[10]. Paired helical filaments (PHFs) and straight filaments (SFs) were found in cases 1 and 3, while only PHFs were in evidence in case 2. The structures of PHFs and SFs were determined to 2.7–3.3-Å resolution and were compared to the previously determined structures of PHFs and SFs from AD[10].

PHFs from the cases with substitution V337M were nearly identical to those of PHFs assembled from wild-type tau in AD. The structures of V337M SFs and AD SFs, which comprise two asymmetrically packed protofilaments A and B with the Alzheimer tau fold, had also similar cross-sections perpendicular to the helical axis. However, unlike SFs of AD, the backbone traces of the protofilaments differed from each other along the helical axis in V337M SFs. In protofilament A, strand β4

of the Alzheimer fold, which comprises residues 336–341, was shifted along the helical axis by about 3 Å compared to protofilament B, which adopted a typical Alzheimer fold (Fig. 2d). Because β4 contains the V337M substitution site, this shift may have resulted from the presence of the mutant residue. The side chain of methionine is bulkier than that of valine but it is also more flexible.

Cryo-EM density maps and the atomic models showed that both wild-type and mutant residues could fit into the density at position 337 (Fig. 2a). In the PHF map of V337M (EMD-19846), the density for the side chain of residue 337 was 50% wider than that of residue V339, when viewed perpendicularly to the filament axis, consistent with it being occupied by a mixture of wild-type (V337) and mutant (M337) residues. Analysis of the sarkosyl-insoluble fractions by mass spectrometry (MS) also showed peptides with either M337 or V337, consistent with the presence of both wild-type and mutant alleles in disease filaments (Extended Data Fig. 2). Immunoblotting of sarkosyl-insoluble tau revealed strong bands of 60, 64 and 68 kDa, as well as a weaker band of 72 kDa, indicating the presence of all six tau isoforms in a hyperphosphorylated state (Fig. 1b). As shown before for sporadic and inherited cases of disease with abundant tau filaments[4,18], all three cases with the *MAPT* mutation encoding V337M showed the presence of variable amounts of high-molecular-weight tau.

We compared the root-mean-square deviation (r.m.s.d.) values of the tau filament structures from the V337M cases to those from other conditions with PHFs and SFs (Extended Data Fig. 3). All values were smaller than 2 Å, indicating the presence of the same folds.

Sarkosyl-insoluble tau from case 3 with substitution V337M contained a previously unknown filament with three-fold symmetry that we named 'triple filament' (TF). We determined the structure of TFs to 2.3-Å resolution (Figs. 1a and 2e). Unlike PHFs and SFs, which are made of two protofilaments, TFs consist of three identical protofilaments, related by $C_3$ symmetry, with each protofilament extending from G273/G304 to E380. Even at 2.3-Å resolution, the side-chain density at the substitution site appeared ambiguous and could accommodate either M337 or V337. A comparison of V337M TF and PHF protofilaments showed that they have similar cross-sections perpendicular to the helical axis but differ by a 3-Å shift of the β4 strand of the TF along the helical axis (Fig. 2f). This shift is like that between V337M SF protofilaments A and B. It is probably essential for TF formation because the N-terminal residues of β4 contribute to the interface between protofilaments,

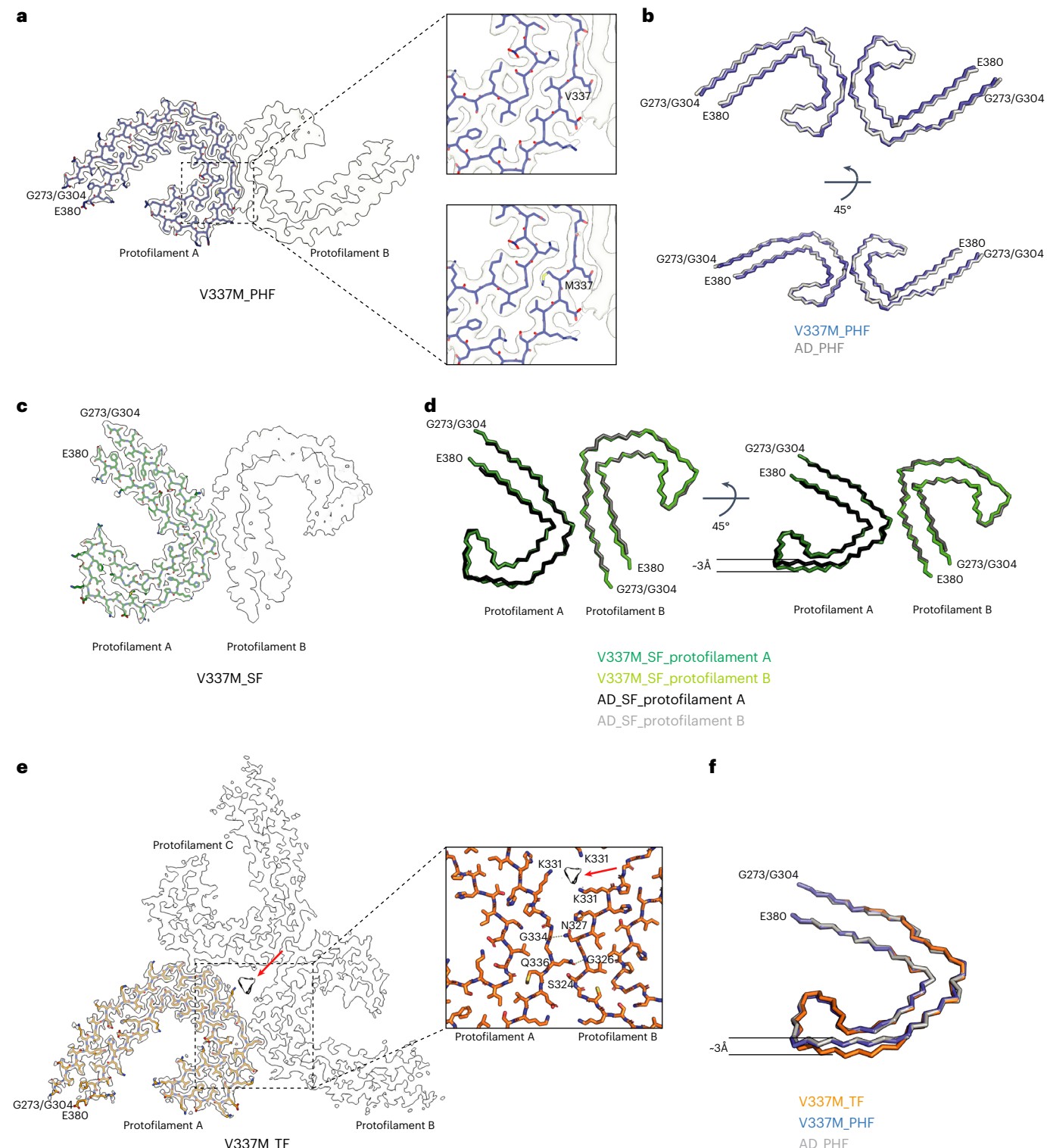

**Fig. 2 | Mutation encoding V337M in *MAPT*: cryo-EM structures of tau filaments. a**, Cryo-EM density map and atomic model of PHF. Two identical protofilaments extend from G273/G304 to E380. Inset, zoomed-in view showing that both wild-type (V) and mutant (M) residues can fit into the density at position 337. **b**, Backbone representation of the overlay of PHFs extracted from the frontal cortex of case 3 with the mutation encoding V337M in *MAPT* (blue) and PHFs extracted from the frontal cortex of an individual with sporadic AD (white; PDB 5O3L). The r.m.s.d. between Cα atoms of the two structures was 0.78 Å. **c**, Cryo-EM density map and atomic model of SF. Two asymmetrically packed protofilaments A and B extend from G273/G304 to E380. **d**, Overlay of SF extracted from the frontal cortex of case 1 with substitution V337M (protofilament A, dark green; protofilament B, light green) and SF extracted from

the frontal cortex of an individual with AD (PDB 5O3T) (protofilament A, black; protofilament B, gray). In protofilament A, strand β4 (residues 336–341) is shifted along the helical axis by 3 Å. Protofilament B adopts the same structure as in AD. **e**, Cryo-EM density map and atomic model of TF. Three identical protofilaments (A, B and C) extend from G273/G304 to E380. An additional nonproteinaceous density at the filament's three-fold axis is labeled with a red arrow. Inset, zoomed-in view showing one of the three identical protofilament interfaces and K331 residues from each protofilament coordinating the additional density. **f**, Overlay of individual protofilaments from TFs with substitution V337M (orange), PHFs with substitution V337M (blue) and PHFs from AD (white), viewed at a 45° angle to the filament axes, as in **d**. The r.m.s.d. between Cα atoms of TFs and V337M tau PHFs was 0.653 Å and that between Cα atoms of TFs and AD PHFs was 0.872 Å.

**Table 1 | Cryo-EM data collection, refinement and validation statistics**

| | V337M case 3 | | | R406W case 2 | In vitro |
|---|---|---|---|---|---|
| **Data collection and processing** | | | | | |
| Microscope | Titan Krios | | | Titan Krios | Titan Krios |
| Voltage (kV) | 300 | | | 300 | 300 |
| Detector | Falcon-4 | | | Falcon-4 | Falcon-4i |
| Magnification | 96,000 | | | 96,000 | |
| Electron exposure (e–/Å²) | 40 | | | 40 | 40 |
| Defocus range (µm) | −1.0 to −2.0 | | | −1.0 to −2.0 | −1.0 to −2.0 |
| Pixel size (Å) | 0.824 | | | 0.824 | 0.727 |
| **Refinement** | PHF (EMD-19846) (PDB 9EO7) | SF (EMD-19849) (PDB 9EO9) | TF (EMD-19852) (PDB 9EOE) | PHF (EMD-19854) (PDB 9EOG) | PHF (EMD-19855) (PDB 9EOH) |
| Initial model used | De novo | De novo | De novo | De novo | De novo |
| Box size (pixel) | 400 | 400 | 400 | 400 | 384 |
| Symmetry imposed | $C_1$ | $C_1$ | $C_3$ | $C_1$ | $C_1$ |
| Initial particle images (no.) | 510,657 | | | 144,450 | 124,071 |
| Final particle images (no.) | 330,371 | 75,617 | 104,579 | 134,218 | 37,720 |
| Map resolution (Å) | 2.8 | 3.3 | 2.3 | 3.0 | 2.8 |
| FSC threshold | 0.143 | 0.143 | 0.143 | 0.143 | 0.143 |
| Helical rise (Å) | 2.37 | 4.76 | 4.76 | 2.37 | 2.37 |
| Helical twist (°) | 179.4 | −1.07 | −0.88 | 179.4 | 179.2 |
| Model resolution (Å) | 2.9 | 3.5 | 2.5 | 3.2 | 2.9 |
| FSC threshold | 0.5 | 0.5 | 0.5 | 0.5 | 0.5 |
| Map sharpening $B$ factor (Å²) | −56 | −83 | −43 | −70 | −42 |
| Model composition | | | | | |
| Nonhydrogen atoms | 4,116 | 4,704 | 3,528 | 3,522 | 3,414 |
| Protein residues | 539 | 616 | 462 | 462 | 444 |
| Ligands | 0 | 0 | 0 | 0 | 0 |
| $B$ factors (Å²) | | | | | |
| Protein | 208.8 | 276.3 | 200.3 | 237.2 | 180.2 |
| r.m.s.d. | | | | | |
| Bond lengths (Å) | 0.0063 | 0.0063 | 0.0063 | 0.0069 | 0.0067 |
| Bond angles (°) | 1.390 | 1.347 | 1.393 | 1.485 | 1.305 |
| **Validation** | | | | | |
| MolProbity score | 1.93 | 2.37 | 2.28 | 2.18 | 1.75 |
| Clashscore | 6.58 | 5.86 | 5.58 | 4.89 | 4.31 |
| Poor rotamers (%) | 1.49 | 5.97 | 4.48 | 4.48 | 0 |
| Ramachandran plot | | | | | |
| Favored (%) | 93.33 | 92.67 | 92.00 | 93.33 | 90.28 |
| Allowed (%) | 6.67 | 7.33 | 8.00 | 6.67 | 9.72 |
| Disallowed (%) | 0 | 0 | 0 | 0 | 0 |

which differ from those of PHFs and SFs. At the interface of TF protofilaments, Q336 from one protofilament intercalates between S324 and N327 of the opposite protofilament and hydrogen bonds with G326. In return, N327 of the opposite protofilament hydrogen bonds with G334 (Fig. 2e). There is a large cavity along the three-fold symmetry axis of the filament, which contains a potentially negatively charged density that is coordinated by residue K331 from each protofilament. It, thus, appears that, like SFs, whose interface contains a nonproteinaceous density between K317 and K321 from both protofilaments, TF assembly may also require external cofactors. Interestingly, case 3 with TFs developed FTD at a younger age than cases 1 and 2 without TFs.

In addition to tau filaments with the Alzheimer fold, TMEM106B filaments were present in case 1 with the V337M substitution (Fig. 1a). This individual died aged 78. The sarkosyl-insoluble fractions from the frontal cortex of cases 2 and 3, who died aged 63 and 58, were devoid of TMEM106B filaments.

### Substitution V337M increases the rate of assembly of recombinant tau (297–391)

We performed in vitro assembly reactions with recombinant proteins to determine whether the V337M substitution in tau (297–391) influences the rate of filament assembly when compared to wild-type

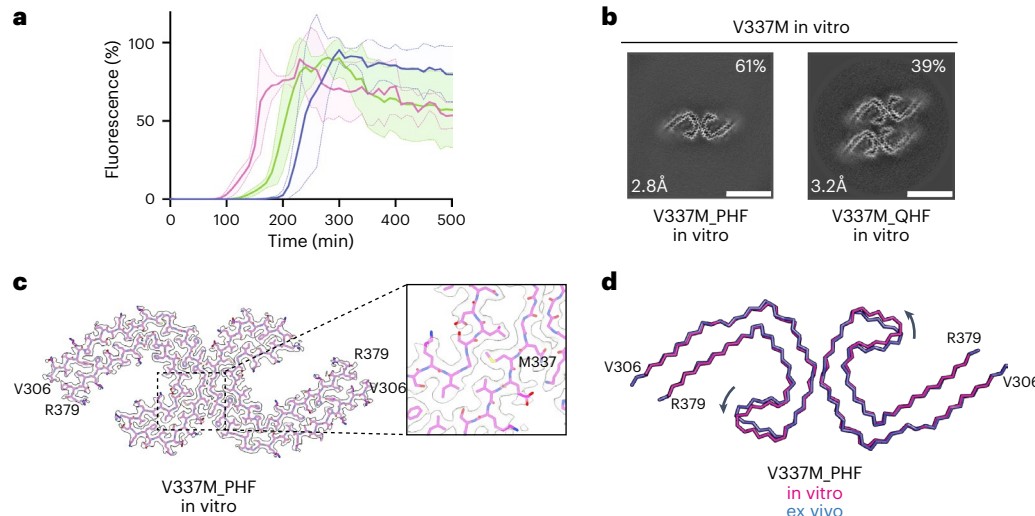

**Fig. 3 | In vitro assembly of V337M tau (297–391). a,** In vitro assembly assay monitored by ThT fluorescence of V337M tau (297–391) (magenta), wild-type tau (297–391) (blue) and a 50:50 mixture of V337M tau (297–391) and wild-type tau (297–391) (green). **b,** Cross-sections through the cryo-EM reconstructions, perpendicular to the helical axis and with a projected thickness of approximately one rung, are shown for assembled V337M tau (297–391). **c,** Cryo-EM density map and atomic model of PHF. Two identical protofilaments extend from V306 to R379. Inset, zoomed-in view showing the mutant methionine at position 337. **d,** Overlay of PHFs assembled from recombinant V337M tau (297–391) (magenta) and extracted from the frontal cortex of an individual with mutation V337M (blue). The r.m.s.d. between Cα atoms was 0.80 Å with a 9° rotation of the β-helix region relative to the rest of the ordered core being the main difference between the two structures.

tau (297–391) (Fig. 3a). With V337M tau (297–391), thioflavin T (ThT) fluorescence started to increase after 90 min and reached a plateau at 180 min. With wild-type tau (297–391), ThT fluorescence began to rise after 200 min and plateaued at 300 min. With a 50:50 mixture of V337M tau (297–391) and tau (297–391), the rate of assembly was intermediate between that of V337M tau (297–391) and tau (297–391). We then proceeded to determine the cryo-EM structures of recombinant V337M tau (297–391) filaments, which revealed the presence of a majority of PHFs and a substantial number of quadruple helical filaments (QHFs) (Fig. 3b,c). The latter, which have been described before[16], are made of two stacked PHFs held together by electrostatic interactions. The cryo-EM density at residue 337 is consistent with a methionine residue (Fig. 3c), demonstrating that this residue at position 337 can give rise to PHFs. These tau filaments exhibited a crossover length of 580 Å, whereas PHFs from human brains have crossover lengths of 700–800 Å. Compared to V337M PHFs from human brains, the recombinant V337M PHFs differed by a slight rotation of the β-helix region with respect to the rest of the ordered core (Fig. 3d).

**Structures of tau filaments from two cases with *MAPT* mutation encoding R406W**

We used two previously unreported cases with the *MAPT* mutation encoding R406W, case 1 from a US family (temporal cortex, parietal cortex and hippocampus) and case 2 from a UK family (frontal cortex, temporal cortex, parietal cortex and hippocampus). According to immunohistochemistry, tau inclusions were present not only in nerve cells and their processes but also in glial cells, chiefly astrocytes (Extended Data Figs. 4–6). In the hippocampus from case 1, many extracellular tau inclusions were present, consistent with the long duration of disease. Overall, neuronal inclusions were more abundant than glial cell inclusions, which ranged from astrocytic plaques to tufted astrocytes, with many intermediates. In some brain regions from R406W case 2, reminiscent of CTE, a subpial tau pathology consisting of thorn-shaped astrocytes was also present at the depths of sulci.

We determined the cryo-EM structures of tau filaments from the temporal cortex, parietal cortex and hippocampus of case 1 and from the frontal, temporal and parietal cortices of case 2 (Figs. 4 and 5). PHFs were present in all samples but we did not observe SFs or TFs. CTE type I filaments were evident in the temporal and parietal cortex from R406W case 2, consistent with the clinicopathological information. The structures of PHFs were determined to resolutions of 3.0–4.2 Å and found to be identical to those of AD PHFs. The ordered core of the R406W tau protofilament extended from G273/G304 to E380.

Immunoblotting of sarkosyl-insoluble tau revealed strong bands of 60, 64 and 68 kDa, as well as a weaker band of 72 kDa, consistent with the presence of all six tau isoforms in a hyperphosphorylated state (Fig. 4b). As shown previously[4], variable amounts of high-molecular-weight tau were also present. According to MS of the sarkosyl-insoluble fractions, we detected only mutant W406 peptides, except in the parietal cortex from case 2, where R406 and W406 peptides were found (Extended Data Fig. 7). In addition to filaments with the Alzheimer and CTE folds, we also observed TMEM106B filaments in the sarkosyl-insoluble fractions from brain regions of both individuals, who died aged 78 and 66, respectively.

## Discussion

We show that mutations encoding V337M and R406W in *MAPT* give rise to the Alzheimer fold. V337 lies inside the ordered core of the Alzheimer fold, whereas R406 lies outside it. Small variations among the observed structures of filaments with the V337M substitution revealed the presence of an adaptable region around the mutation site that explains the accommodation of the mutant methionine without disruption of the overall fold.

Using MS, we found both wild-type and mutant tau in the core of filaments extracted from the frontal cortex of three individuals with substitution V337M, indicating that the Alzheimer tau fold can accommodate V337 and M337. Tau filaments extracted from the parietal cortex of participant 1 with mutation R406W also contained both wild-type and mutant proteins, while filaments from the cerebral cortex and hippocampus of participant 2 appeared to have only mutant tau in the filaments. The former finding is in line with a study that reported the presence of wild-type and mutant forms of tau in the filaments from cases with substitution R406W (ref. 32).

Our results are consistent with positron-emission tomography scanning using ¹⁸F-labeled flortaucipir that showed binding to tau inclusions in persons with *MAPT* mutations encoding V337M and

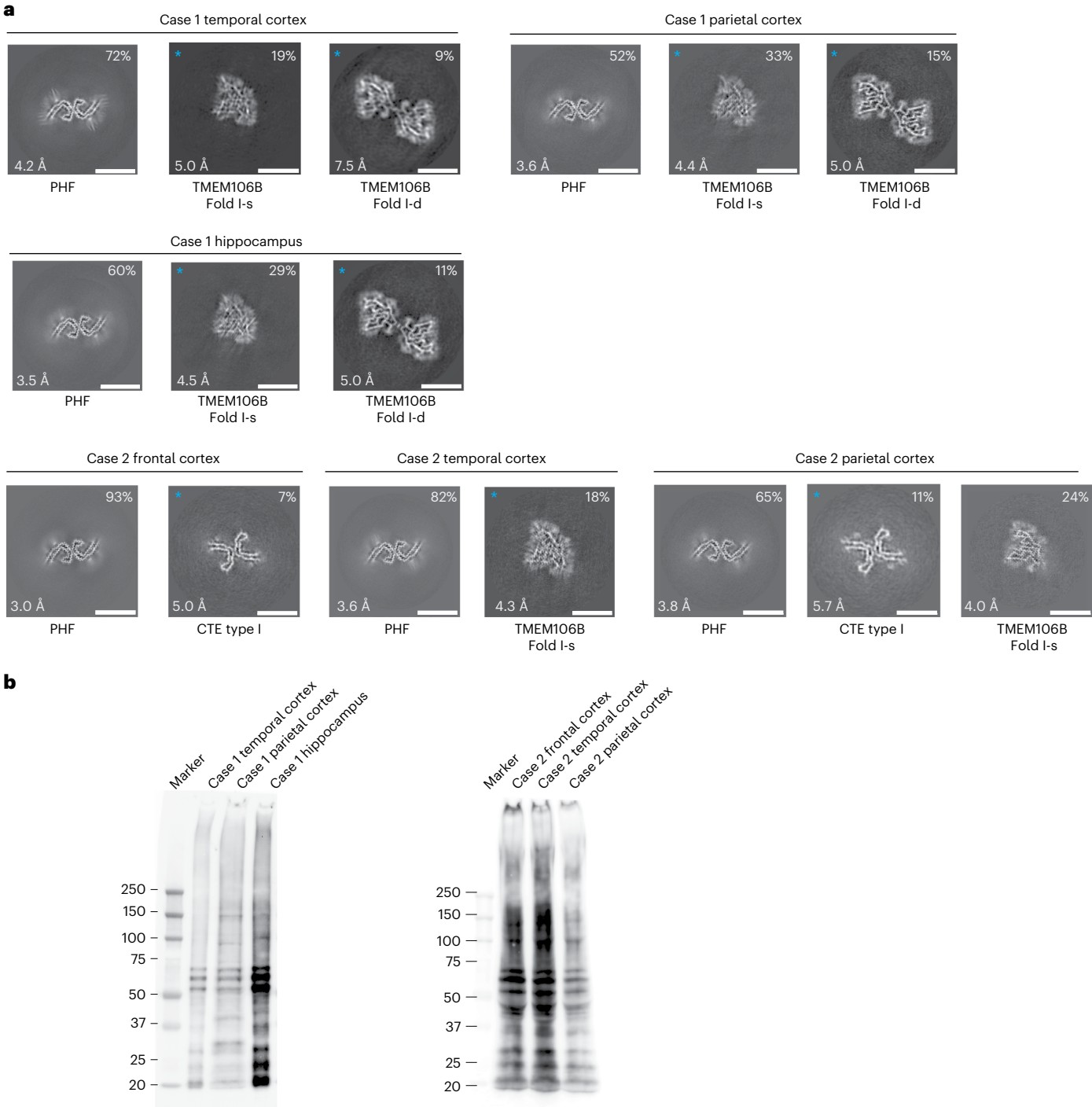

**Fig. 4 | Mutation encoding R406W in *MAPT*: cryo-EM cross-sections of tau filaments and immunoblotting. a**, Cross-sections through the EM reconstructions, perpendicular to the helical axis and with a projected thickness of approximately one rung, are shown for the temporal cortex, parietal cortex and hippocampus of case 1 and for the frontal, temporal and parietal cortices of case 2. Resolutions (in Å) and percentages of filament types are indicated at the bottom left and top right, respectively. Scale bar, 10 nm. **b**, Immunoblotting of sarkosyl-insoluble tau from the temporal cortex, parietal cortex and hippocampus of case 1 with substitution R406W and from the frontal, temporal and parietal cortices of case 2 with substitution R406W. Phosphorylation-independent anti-tau antibody BR134 was used.

R406W (refs. [21,27,33,34]). [18]F-labeled flortaucipir retention has also been shown to be associated with the tau pathology of AD[35] and some prion protein amyloidoses[36]. Like persons with AD, R406W mutant carriers had elevated levels of tau in cerebrospinal fluid, as measured by the antibody MTBR-tau243 (ref. [37]).

The observation that cases of FTDP-17 can have the same tau filament fold as cases of AD further illustrates the fact that even though specific tau folds characterize distinct diseases, the same fold can be associated with clinically different conditions. Mutations in *MAPT* do not give rise to familial AD. We showed previously that other cases of FTD and parkinsonism linked to chromosome 17 (FTDP-17) adopted the Pick[5] or the AGD[6] fold, depending on the relative overexpression of 3R or 4R tau.

Mutations encoding V337M and R406W in *MAPT* led to the formation of extensive neuronal tau pathology in the form of intracellular inclusions that were reactive with antibodies RD3 and RD4, which are

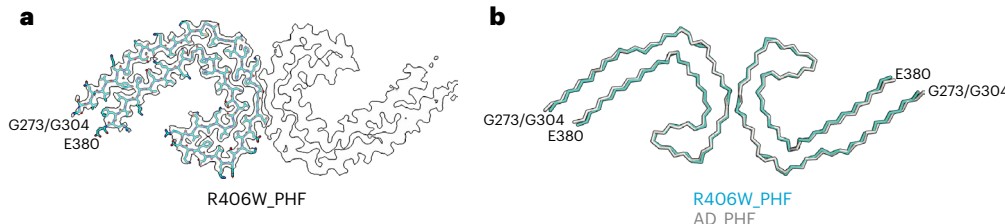

**Fig. 5 | Mutation encoding R406W in *MAPT*: cryo-EM structures of tau filaments. a,** Cryo-EM density and atomic model of PHF from the frontal cortex of case 2. Two identical protofilaments extend from G273/G304 to E380.

**b,** Overlay of PHFs extracted from the frontal cortex of case 2 (blue) and the frontal cortex of a case of sporadic AD (black). The r.m.s.d. between Cα atoms of the two structures was 0.78 Å.

specific for 3R and 4R tau, respectively. In agreement with previous work[18,24], immunoblotting of sarkosyl-insoluble fractions showed a pattern of tau bands typical of all six isoforms in a hyperphosphorylated state.

In case 1 with substitution R406W, many extracellular tau inclusions (ghost tangles) were present in the hippocampus, reflecting the long duration of disease. A tangle becomes extracellular after the neuron that contained it dies. Whereas intracellular inclusions are made of full-length tau, ghost tangles progressively lose their fuzzy coat and consist mainly of the ordered filament core (R3, R4 and 10–12 amino acids after R4). These sequences are common to 3R and 4R tau isoforms. Extracellular tau inclusions can be abundant in cases with Alzheimer and CTE tau folds[38,39] and their insolubility has been attributed to extensive crosslinks[40]. They are much less frequent in cases with the folds of 3R and 4R tauopathies, indicating a link between filaments made of all six tau isoforms and the formation of ghost tangles.

There were also astrocytic tau inclusions in the cases with substitution R406W, suggesting that both nerve cell and glial cell inclusions contained the Alzheimer fold. Previously, shared tau folds between nerve cells and glial cells were reported for Pick's disease, progressive supranuclear palsy, corticobasal degeneration and globular glial tauopathy[5,6,41].

It remains to be determined how substitutions V337M and R406W in *MAPT* cause FTDP-17. Previous studies showed that they lead to small reductions in the ability of recombinant tau to interact with microtubules[42,43]. This partial loss of function may be necessary for the assembly into filaments. It has been shown that substitutions V337M and R406W do not notably influence the heparin-induced assembly of full-length tau[44]. However, the structures of heparin-induced tau filaments differ from those of AD[45]. By contrast, recombinant tau (297–391) gives rise to PHFs[16]. Because substitution V337M is inside the filament core, we assembled V337M tau (297–391); PHFs and QHFs formed, with a marked increase in the rate of filament assembly compared to wild-type tau (297–391). An intermediate increase in the assembly rate was observed when 50:50 mixtures of V337M tau (297–391) and tau (297–391) were used (the mutation encoding V337M is dominantly inherited). It remains to be seen whether the formation of QHFs contributed to this effect. These findings suggest that substitution V337M has a direct effect on tau filament assembly and demonstrate the usefulness of V337M tau (297–391) for increasing filament formation in experimental studies.

In conclusion, mutations in *MAPT* can give rise to cases of FTDP-17 that resemble sporadic tauopathies, including Pick's disease and AGD. Thus, for mutations with primary effects at the splicing level, the Pick fold forms when filaments are made of wild-type 3R tau[5], whereas the AGD fold forms when filaments are made of wild-type 4R tau[6]. Here, we show that missense mutations encoding V337M and R406W, which give rise to an amnestic phenotype resembling that of AD and biochemical changes resembling those in AD, result in the formation of the Alzheimer tau fold. It follows that cases of FTDP-17 that are caused

by mutations in *MAPT* can be divided into distinct subgroups whose filament structures can be defined by cryo-EM.

## Online content

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

## Methods

### Ethics
For the V337M *MAPT* cases, informed consent for brain donation was obtained from the legal next of kin according to protocols approved by the University of Washington (UWA) Institutional Review Board that conform to the provisions of the Declaration of Helsinki and preserve donor anonymity. For R406W *MAPT* case 1, research protocols for the Indiana AD Research Center were approved by the Indiana University Institutional Review Board (protocol no. 1011003338; initial approval date: April 22, 1991, current expiration date: February 5, 2025). Brain tissue from R406W *MAPT* case 2 was donated to the University College London Queen Square Brain Bank with informed consent and the study was approved by the National Health Services Health Research Authority Ethics Committee, London-Central (reference no. 23/LO/0044). Genomic DNA from the V337M and R406W cases was extracted from postmortem brain tissues. The cryo-EM study was approved by the Cambridgeshire Research Ethics committee (09/HO308/163).

### Cases with *MAPT* mutation encoding V337M (Seattle family A)
We used the frontal cortex from three previously described cases of Seattle family A with the mutation encoding V337M in *MAPT*: case III-1, case IV-4 (ref. [17]) and case IV-60 (ref. [20]). We also used the hippocampus from cases III-1 and IV-4. All three individuals developed a variety of symptoms, some psychiatric, consistent with a diagnosis of behavioral-variant FTD. Case 1 (III-1, UWA 63) was a female who died aged 78. At age 52, she became uncooperative, hostile, suspicious and withdrawn. She also developed progressive memory loss. Case 2 (IV-4, UWA 271) was a female who died aged 63, following an 11-year history of FTD. She was the daughter of case 1 and presented with antisocial and impulsive behaviors, which were followed by apathy, loss of language and dementia. Case 3 (IV-60, UWA 578) was a male who died aged 58 following a 16-year history of FTD. He lost his job because of poor performance, was vague and restless and behaved inappropriately. Early on, he had mild memory problems and deficient executive function. His condition slowly progressed to dementia requiring hospitalization.

### Case with *MAPT* mutation encoding R406W (US family)
We used the temporal cortex, parietal cortex and hippocampus from a female with the mutation encoding R406W in *MAPT* who died aged 78, after a 29-year history of personality changes and cognitive impairment. The clinical diagnosis was AD. Genetic or neuropathological information on the parents was not available but the mother had been diagnosed with AD. In addition to the proband, she had four other children (three females and one male), who all developed cognitive impairment in mid-life. They were diagnosed with AD (three females) or FTDP-17 (male). The symptoms of the proband were dominated by progressive dementia and personality changes characterized by an anxiety disorder.

### Case with *MAPT* mutation encoding R406W (UK family)
We used frontal, temporal and parietal cortices from a male with the mutation encoding R406W in *MAPT* who died aged 66 after a 9-year history of FTD. Both parents died without known FTD before the age of 61. At least one sibling developed FTD. The initial symptoms were episodic memory impairment with subsequent executive dysfunction and personality changes characterized by impulsivity and inappropriate behavior. Magnetic resonance imaging showed severe bilateral frontal lobe and medial temporal lobe atrophy that was more severe on the left side. This individual worked as an electrician until the age of 55 and had a history of alcohol abuse. In his youth, he had played soccer for several years.

### DNA sequencing
Genomic DNA was extracted from human brains. Standard amplification reactions were performed with 50 ng of genomic DNA, followed by DNA sequencing of exons 1 and 9–13 of *MAPT* with adjoining intronic sequences, as described previously[46].

### Filament extraction from human brains
Sarkosyl-insoluble material was extracted from the frontal cortex of cases 1–3 with the mutation encoding V337M and the frontal, temporal and parietal cortices of two cases with the mutation encoding R406W in *MAPT*, as described previously[47]. The hippocampus from case 2 was also used. Tissues were homogenized in 20 vol% (w/v) buffer A (10 mM Tris-HCl pH 7.4, 0.8 M NaCl, 10% sucrose and 1 mM EGTA), brought to 2% sarkosyl and incubated at 37 °C for 30 min. The samples were centrifuged at 7,000$g$ for 10 min, followed by spinning of the supernatants at 100,000$g$ for 60 min. The pellets were resuspended in buffer A (1 ml g$^{-1}$ tissue) and centrifuged at 9,500$g$ for 10 min. The supernatants were diluted threefold in buffer B (50 mM Tris-HCl pH 7.5, 0.15 M NaCl, 10% sucrose and 0.2% sarkosyl), followed by a 60-min spin at 100,000$g$. For cryo-EM, the pellets were resuspended in 100 µl g$^{-1}$ buffer C (20 mM Tris-HCl pH 7.4 and 100 mM NaCl).

### Immunoblotting and histology
For immunoblotting, samples were resolved on 4–12% Bis-Tris gels (NuPage) and the primary antibody (BR134; 1:1,000)[1] was diluted in PBS plus 0.1% Tween 20 and 5% nonfat dry milk. Histology and immunohistochemistry were carried out as described[46]. Some sections (8 µm) were counterstained with hematoxylin. The primary antibodies were AT8 (Thermo Fisher Scientific; 1:1,000 or 1:300), RD3 (Sigma-Millipore, 1:3,000) RD4 (Sigma-Millipore 1:100) and anti-4R (Cosmo Bio 1:400).

### Expression and purification of recombinant tau (297–391) with and without V337M substitution
Tau (297–391) with the V337M substitution was made using in vivo assembly[48]. Reverse and forward primers were designed to share 15 nucleotides of the homologous region and 15–30 nucleotides for annealing to the template. Expression of tau (297–391) was carried out in *Escherichia coli* BL21(DE3)-gold cells (Agilent Technologies), as described previously[49]. One plate of cells was resuspended in 1 L of 2× TY (tryptone yeast) supplemented with 2.5 mM MgSO$_4$ and 100 mg L$^{-1}$ ampicillin and cells were grown to an optical density of 0.8 at 600 nm. They were induced by the addition of 1 mM IPTG for 4 h at 37 °C, collected by centrifugation (4,000$g$ for 30 min at 4 °C) and flash-frozen. The pellets were resuspended in washing buffer at room temperature (50 mM MES pH 6.5, 10 mM EDTA, 10 mM DTT and 0.1 mM PMSF) and cells were lysed by sonication (90% amplitude using a Sonics VCX-750 Vibracell ultrasonic processor for 4 min with 3 s on and 6 s off) at 4 °C. The lysed cells were centrifuged at 20,000$g$ for 35 min at 4 °C, filtered through 0.45-µm-cutoff filters and loaded onto a HiTrap CaptoS 5-ml column at 4 °C (GE Healthcare). The column was washed with ten volumes of buffer, followed by elution through a gradient of washing buffer containing 0–1 M NaCl. Fractions of 3.5 ml were collected and analyzed by SDS–PAGE (4–20% Tris–glycine gels). Protein-containing fractions were pooled and precipitated using 0.3 g ml$^{-1}$ ammonium sulfate and left on a rocker for 30 min at 4 °C. The solution was then centrifuged at 20,000$g$ for 35 min at 4 °C and resuspended in 2 ml of 10 mM potassium phosphate buffer pH 7.2 containing 10 mM DTT and loaded onto a 16/600 75-pg size-exclusion column. Fractions were analyzed by SDS–PAGE and protein-containing fractions were pooled and concentrated at 4 °C to 20 mg ml$^{-1}$ using molecular weight concentrators with a cutoff filter of 3 kDa. Purified protein samples were flash-frozen in 50–100-µl aliquots. Protein concentrations were determined using a NanoDrop 2000 (Thermo Fisher Scientific).

### Filament assembly of recombinant tau (297–391) with and without V337M substitution
Before assembly, proteins and buffers were filtered through sterile 0.22 µM Eppendorf filters. A solution of 6 mg ml$^{-1}$ wild-type tau

(297–391) or V337M mutant tau (297–391) was prepared at room temperature in 50 mM potassium phosphate pH 7.2, 10 mM DTT and 2 µM ThT. An additional set of reactions was prepared without ThT for cryo-EM analysis. Then, 30-µl aliquots were dispensed in a 384-well plate (company) that was sealed and placed in a Fluostar Omega (BMG Labtech) plate reader. Assembly was carried out using orbital shaking (200 rpm) at 37 °C for 12 h.

## MS

Sarkosyl-insoluble pellets were resuspended in 200 µl of hexafluoroisopropanol. Following a 3-min sonication at 50% amplitude (QSonica), they were incubated at 37 °C for 2 h and centrifuged at 100,000$g$ for 15 min, before being dried by vacuum centrifugation. Protein samples resuspended in 4 M urea and 50 mM ammonium bicarbonate were reduced with 5 mM DTT at 37 °C for 40 min and alkylated with 10 mM chloroacetamide for 30 min. V337M samples were digested with LysC (Promega) for 4 h followed by trypsin after dilution of urea to 1.5 M. For R406W samples, urea was diluted to 1.0 M and incubated with AspN (Promega) overnight at 30 °C. Digestion was stopped by the addition of formic acid to a final concentration of 0.5%, followed by centrifugation at 16,000$g$ for 5 min. The supernatants were desalted and fractionated using homemade C18 stage tips (3 M Empore) packed with Poros Oligo R3 (Thermo Fisher Scientific) resin. Bound peptides were eluted stepwise with increasing acetonitrile in 10 mM ammonium bicarbonate and partially dried in a SpeedVac (Savant). Samples were analyzed by liquid chromatography (LC)–MS/MS using a Q Exactive Plus hybrid quadrupole Orbitrap MS instrument (Thermo Fisher Scientific) coupled online to a fully automated Ultimate 3,000 rapid separation LC nano system (Thermo Fisher Scientific). LC–MS/MS data were searched against the human reviewed database (UniProt, downloaded 2023), using the Mascot search engine (Matrix Science, version 2.80). Scaffold (version 4, Proteome Software) was used to validate MS/MS-based peptide and protein identifications.

## Cryo-EM

Cryo-EM grids (Quantifoil 1.2/1.3, 300 mesh) were glow-discharged for 1 min using an Edwards (S150B) sputter coater. Then, 3 µl of the sarkosyl-insoluble fractions or recombinant Tau assemblies were applied to the glow-discharged grids, followed by blotting with filter paper and plunge freezing into liquid ethane using a Vitrobot Mark IV (Thermo Fisher Scientific) at 4 °C and 100% humidity. Cryo-EM images were acquired on a Titan Krios G2 or G4 microscope (Thermo Fisher Scientific) operated at 300 kV and equipped with a Falcon-4 or a Falcon-4i direct electron detector using EPU software (Thermo Fisher Scientific). Images were recorded for 2 s in electron event representation format[50], with a total dose of 40 electrons per A² and a pixel size of 0.824 Å (Falcon-4) or 0.727 Å (Falcon-4i). Further details can be found in Table 1 and Extended Data Figs. 8–10.

## Data processing

Datasets were processed in RELION using standard helical reconstruction[51,52]. Video frames were gain-corrected, aligned and dose-weighted using RELION's own motion correction program[53]. The contrast transfer function (CTF) was estimated using CTFFIND4.1 (ref. 54). Filaments were picked manually and segments were extracted with a box size of 1,024 pixels, before downsizing to 256 pixels. Reference-free two-dimensional (2D) classification was carried out and selected class averages were re-extracted using a box size of 400 pixels. Initial models were generated de novo from 2D class average images using relion_helix_inimodel2d (ref. 55). Three-dimensional refinements were performed in RELION-4.0 and the helical twist and rise were refined using local searches. Bayesian polishing and CTF refinement were used to further improve resolutions[56]. The final maps were sharpened using postprocessing procedures in RELION-4.0 and resolution estimates were calculated on the basis of the Fourier shell correlation (FSC) between two independently refined half-maps at 0.143 (Extended Data Fig. 10)[57]. We used relion_helix_toolbox to impose helical symmetry on the postprocessing maps.

## Model building and refinement

Atomic models were built manually using Coot[58] on the basis of published structures (PHF, Protein Data Bank (PDB) 5O3L; SF, PDB 5O3T)[10]. Model refinements were performed using ISOLDE[59], Servalcat[60] and REFMAC5 (refs. 61,62). Models were validated with MolProbity[63]. Figures were prepared with ChimeraX[64] and PyMOL (Schrödinger). When multiple maps of the same filament type were resolved, atomic modeling and database submission were only performed for the map with the highest resolution.

## Reporting summary

Further information on research design is available in the Nature Portfolio Reporting Summary linked to this article.

## Data availability

Cryo-EM maps were deposited to the EM Data Bank under the following accession codes: EMD-19846, EMD-19849, EMD-19852, EMD-19854 and EMD-19855. Corresponding refined atomic models were deposited to the PDB under the following accession codes: 9EO7, 9EO9, 9EOE, 9EOG and 9EOH. Please address requests for materials to the corresponding authors. Source data are provided with this paper.

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

## Acknowledgements

We thank the participants' families for donating brain tissues, T. Darling, I. Clayson and J. Grimmett for help with high-performance computing and the EM facility of the Medical Research Council (MRC) Laboratory of Molecular Biology for help with cryo-EM data acquisition. We are grateful to R. Richardson, N. Maynard, M. Jacobsen and B. Glazier for help with histology and immunohistochemistry. This work was supported by the MRC, as part of UK Research and Innovation (MC_UP_A025_1013 to S.H.W.S. and MC_1051284291 to M.G.). It was also supported by the US National Institutes of Health (P30-AG010133, R01-AG080001 and RF1-AG071177, to B.G.) and the Department of Pathology and Laboratory Medicine, Indiana University School of Medicine (to B.G.). The Queen Square Brain Bank is supported by the Rita Lila Weston Institute for Neurological Studies.

## Author contributions

M.B., C.L., J.R.M., P.W.C., Z.J., T.D.B. and B.G. identified participants and performed neuropathology and DNA sequencing. C.Q. performed immunoblot analysis. C.Q., S.P.-C and C.F. performed MS. C.Q. and S.L. collected cryo-EM data. C.Q., S.L., A.G.M. and S.H.W.S. analyzed cryo-EM data. S.H.W.S. and M.G. supervised the project. All authors contributed to the writing of the manuscript.

## Competing interests

The authors declare no competing interests.

## Additional information

**Extended data** is available for this paper at https://doi.org/10.1038/s41594-025-01498-5.

**Correspondence and requests for materials** should be addressed to Sjors H. W. Scheres or Michel Goedert.

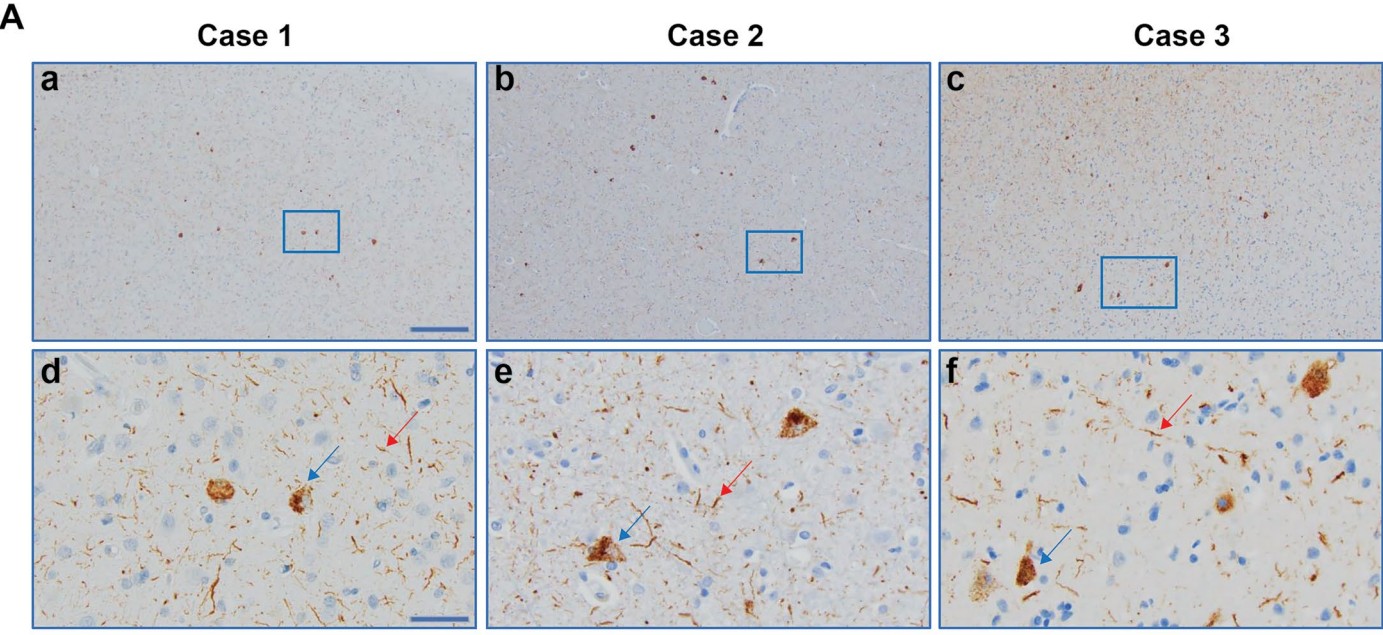

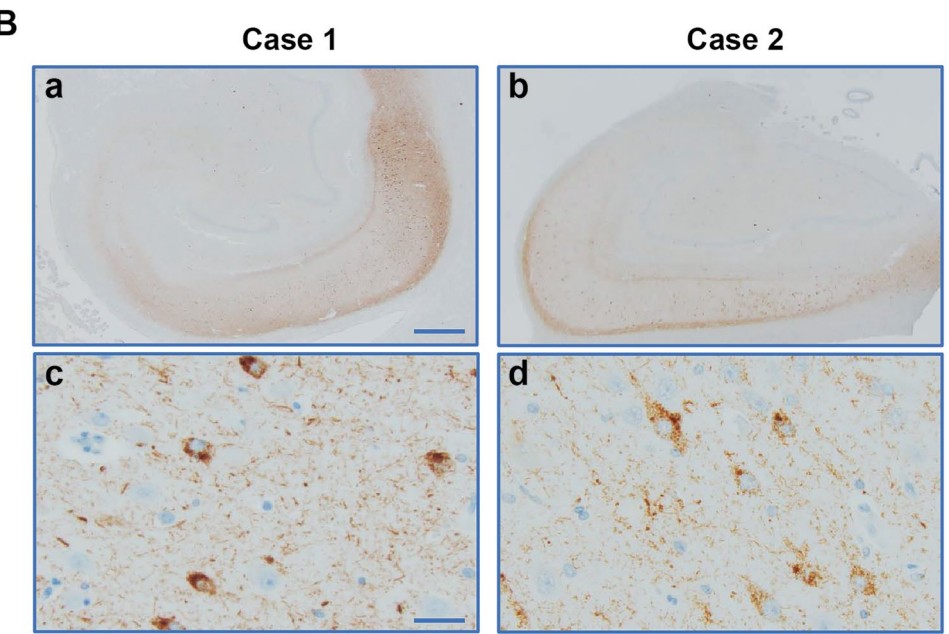

**Extended Data Fig. 1 | *MAPT* mutation encoding V337M: immunohistochemical localization of tau inclusions in frontal cortex and hippocampus. A. a-f,** Tau pathology in grey matter of frontal cortex. Higher magnifications of tissue areas within the insets in **a-c** are shown in **d-f**. Intraneuronal inclusions (blue arrows) and neuropil threads (red arrows) are in evidence. Antibody: AT8. Scale bars, 200 μm (**a-c**); 40 μm (**d-f**). **B. a,b,** Low magnification views of tau pathology in grey matter of hippocampus. **c,d,** Neurofibrillary tangles and neuropil threads in the pyramidal cell layer. Antibody: AT8. Scale bars, 1,000 μm (**a,b**) and 40 μm (**c,d**).

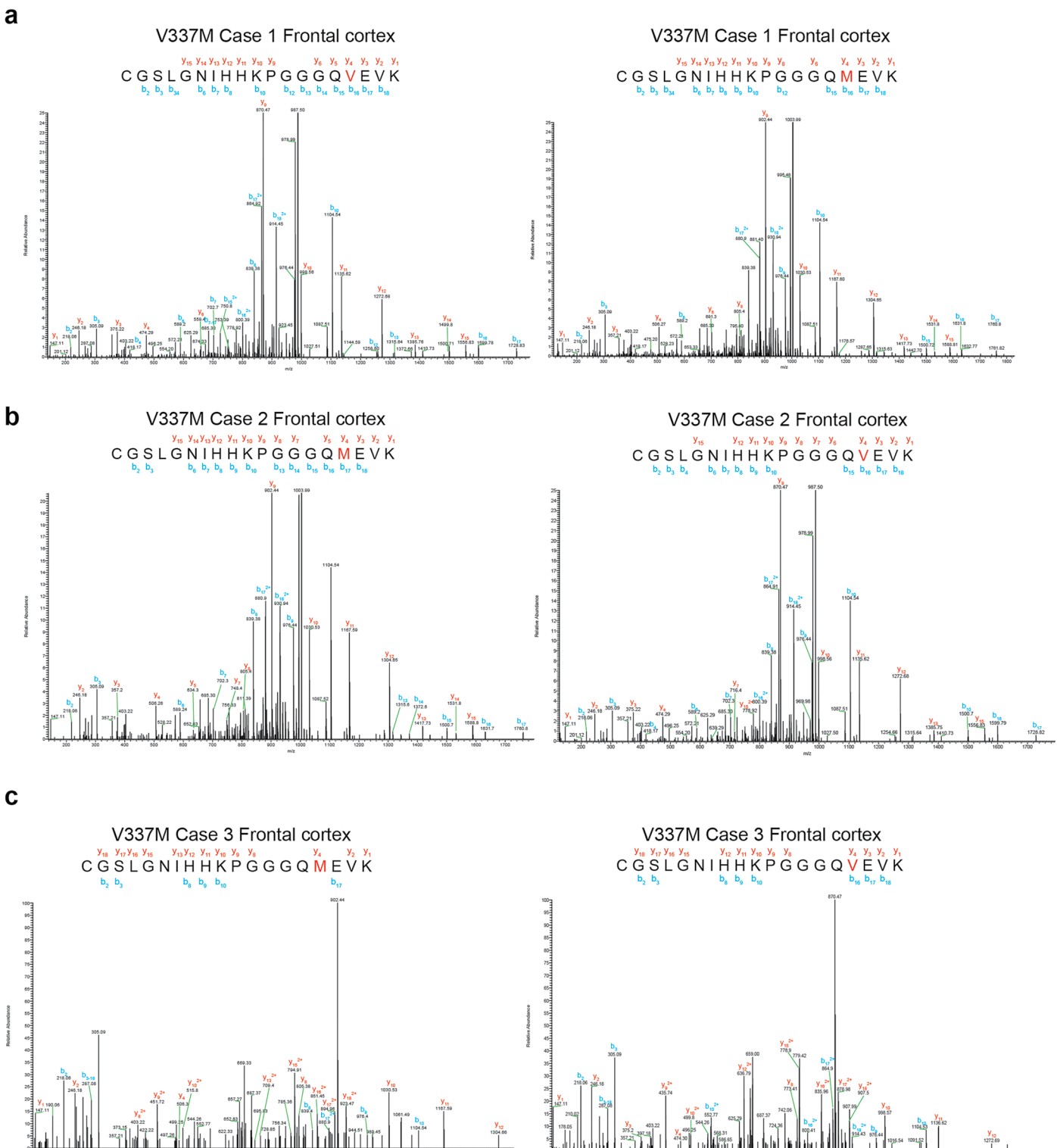

**Extended Data Fig. 2 | Mass spectrometry of tau from the sarkosyl-insoluble fractions of cases with *MAPT* mutation encoding V337M.** MALDI mass spectra of the frontal cortex from cases 1–3 (**a-c**). Wild-type (V337) and mutant (M337) tau peptides were detected.

**a**    **b**

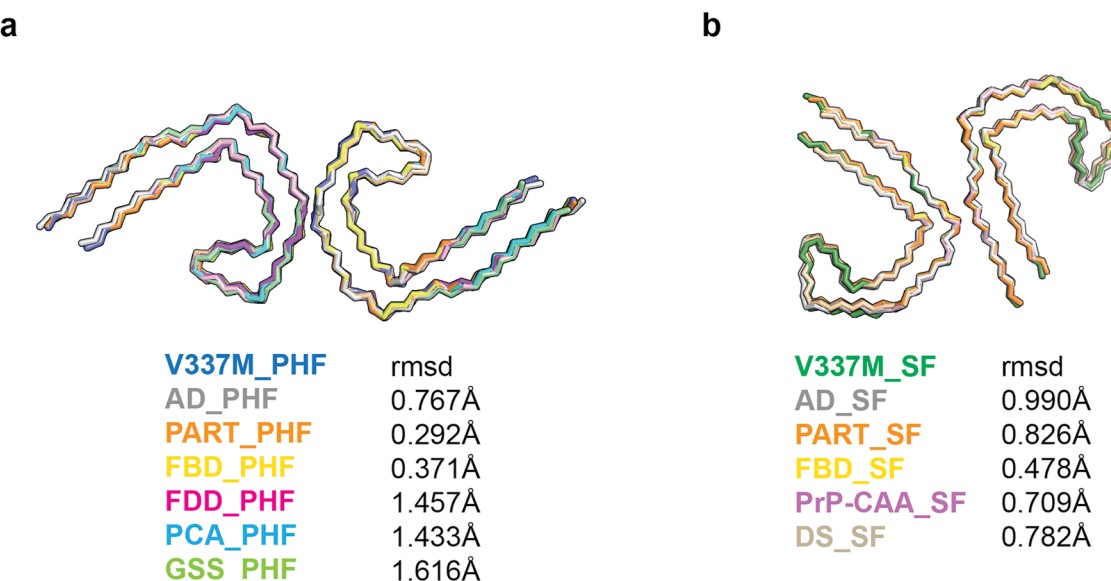

| V337M_PHF | rmsd |
|---|---|
| AD_PHF | 0.767Å |
| PART_PHF | 0.292Å |
| FBD_PHF | 0.371Å |
| FDD_PHF | 1.457Å |
| PCA_PHF | 1.433Å |
| GSS_PHF | 1.616Å |
| PrP-CAA_PHF | 1.639Å |
| DS_PHF | 0.490Å |

| V337M_SF | rmsd |
|---|---|
| AD_SF | 0.990Å |
| PART_SF | 0.826Å |
| FBD_SF | 0.478Å |
| PrP-CAA_SF | 0.709Å |
| DS_SF | 0.782Å |

**Extended Data Fig. 3 | Comparison of V337M tau filament structures to those from other conditions.** Tau paired helical filaments (PHFs) (**a**) and straight filaments (SFs) (**b**) from the frontal cortex of individuals with missense mutation V337M in MAPT were compared to those of PHFs and SFs from Alzheimer's disease (AD), primary age-related tauopathy (PART), familial British dementia (FBD), familial Danish dementia (FDD), posterior cortical atrophy (PCA), Gerstmann-Sträussler-Scheinker disease (GSS), cerebral amyloid angiopathy with prion protein deposits (PrP-CAA) and Down's syndrome (DS).

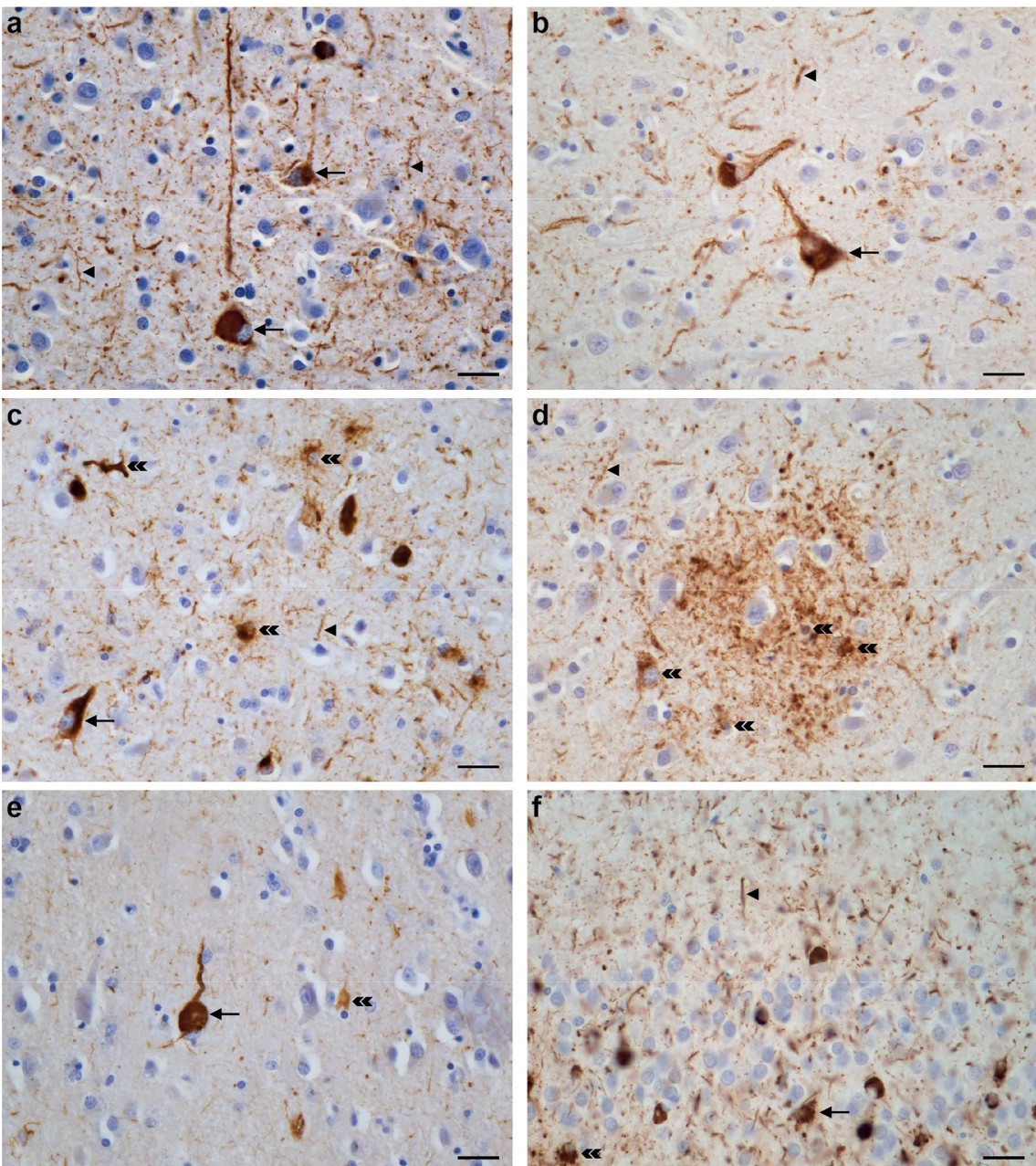

**Extended Data Fig. 4 | *MAPT* mutation encoding R406W: immunohistochemical localization of tau in temporal cortex, parietal cortex and hippocampus of case 1. a,c,e**, Tau-immunopositive nerve cell bodies (arrows) and neuropil threads (arrowheads) are shown in temporal cortex; **b,d**, parietal cortex; **f**, hippocampus. **c,d,e,f**, Labelled astrocytes (double arrowheads). Panel (**d**) shows plaques composed of numerous threads, corresponding probably to the processes of neurons and astrocytes. Antibodies: AT8 (**a,b,d,f**); RD4 (**c**); RD3 (**e**). Scale bar, 25 μm.

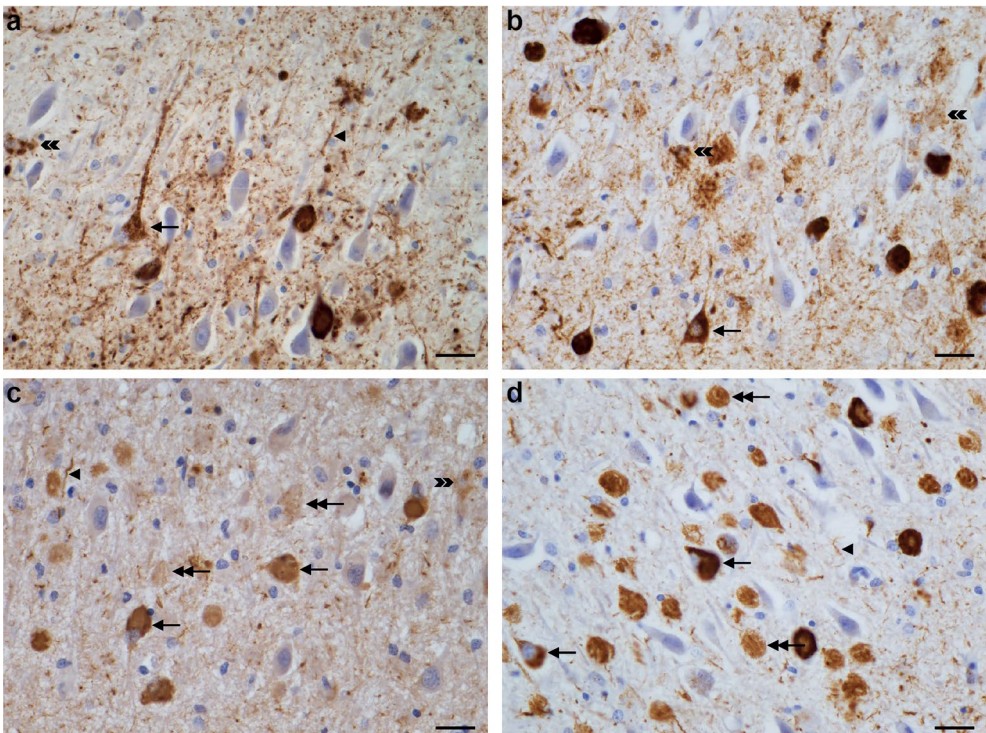

**Extended Data Fig. 5 | *MAPT* mutation encoding R406W: immunohistochemical localization of tau inclusions in the hippocampus of case 1.** The pyramidal layer is shown. Tau-immunopositive intracellular neuronal inclusions (single headed arrows) and extracellular ghost inclusions (double headed arrows), as well as neuropil threads (single arrowheads) and astrocytic inclusions (double arrowheads) are indicated. Antibodies: AT8 (**a**), RD4 (**b**), anti-4R (**c**), RD3 (**d**). Scale bar, 25 μm.

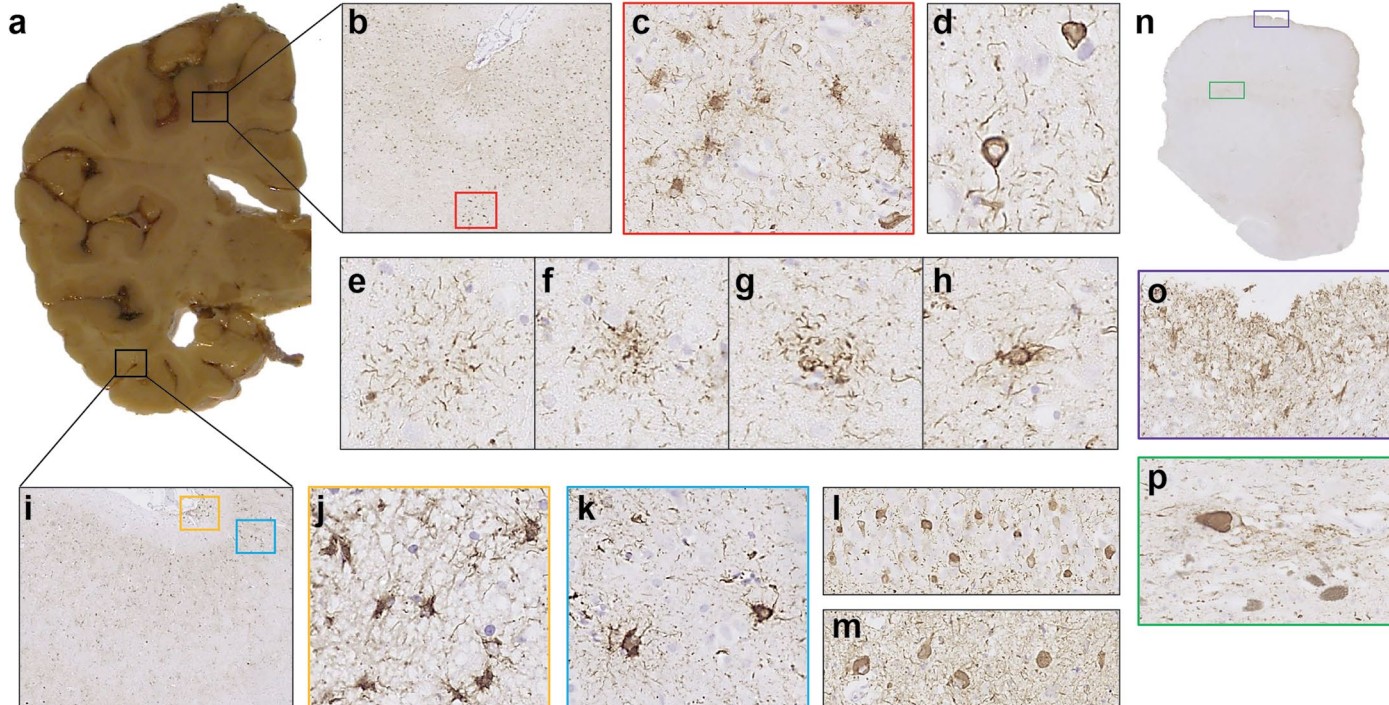

**Extended Data Fig. 6 | *MAPT* mutation encoding R406W: immunohistochemical localization of tau inclusions in case 2. a**, Mild atrophy of the frontal cortex, severe atrophy of the temporal cortex and underlying white matter, with marked reduction in bulk of the hippocampus. Anterior and temporal horns of the lateral ventricle are dilated; **b**, tau pathology in the anterior frontal cortex; **c**, some stained cells resemble tufted or thorn-shaped astrocytes; **d**, abundant neuronal inclusions and neuropil threads; **e**, astrocytic plaque; **f,g**, structures in-between astrocytic plaques and tufted astrocytes; **h**, tufted astrocyte; **i,k**, Tau pathology in the lateral temporal cortex was similar to that in the anterior frontal cortex; **l**, CA4 region of the hippocampus; m, dentate gyrus; **j**, subpial astrocytic tau pathology at the depth of a sulcus in the lateral temporal cortex. **n**, Low-power view of the midbrain; **o**, subpial astrocytic tau pathology; **p**, neuronal tau staining in the substantia nigra. AT8 antibody. Scale bar: **b**, 750 μm; **c,k**, 70 μm; **d**, 40 μm; **e,f,g,h**, 30 μm; **i**, 670 μm; **j**, 50 μm; **l**. 100 μm; **m**, 110 μm; **n**, 5.5 μm; **o,p**, 90 μm.

**a**

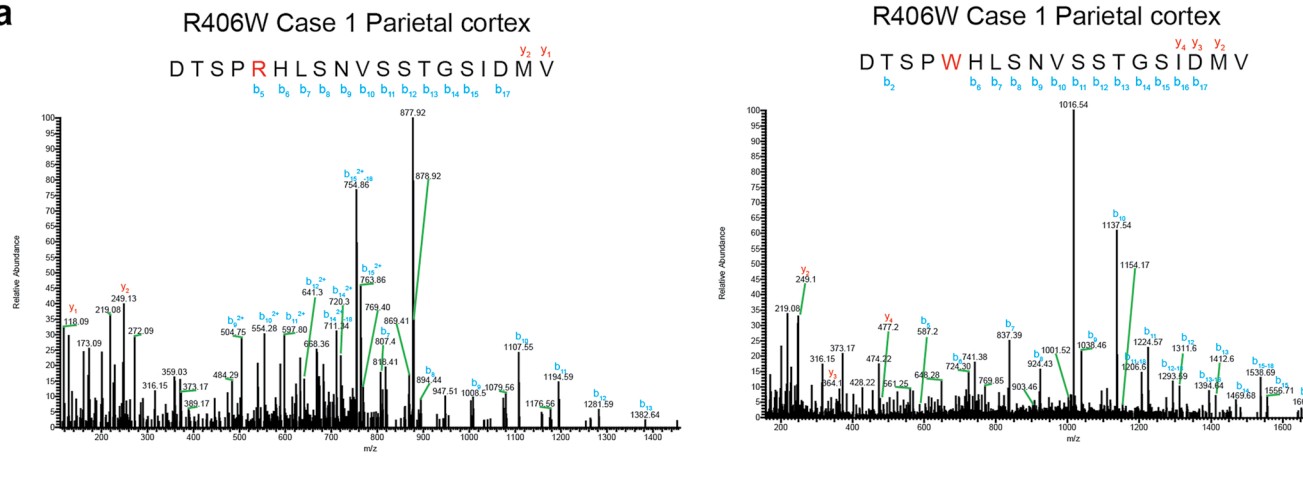

**b**

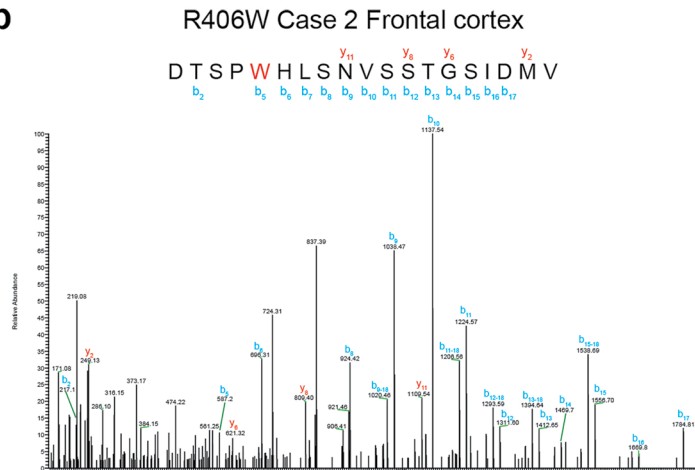

**Extended Data Fig. 7 | Representative mass spectrometry of tau from the sarkosyl-insoluble fractions of cases with *MAPT* mutation encoding R406W.** MALDI mass spectra of parietal cortex from case 1 (**a**) and frontal cortex from case 2 (**b**). Wild-type (R406) and mutant (W406) peptides were detected in parietal cortex from case 1. Only mutant peptides (W406) were detected in frontal cortex from case 2. Similarly, only mutant peptides were detected in temporal cortex and hippocampus from case 1, as well as in temporal cortex, parietal cortex and hippocampus from case 2.

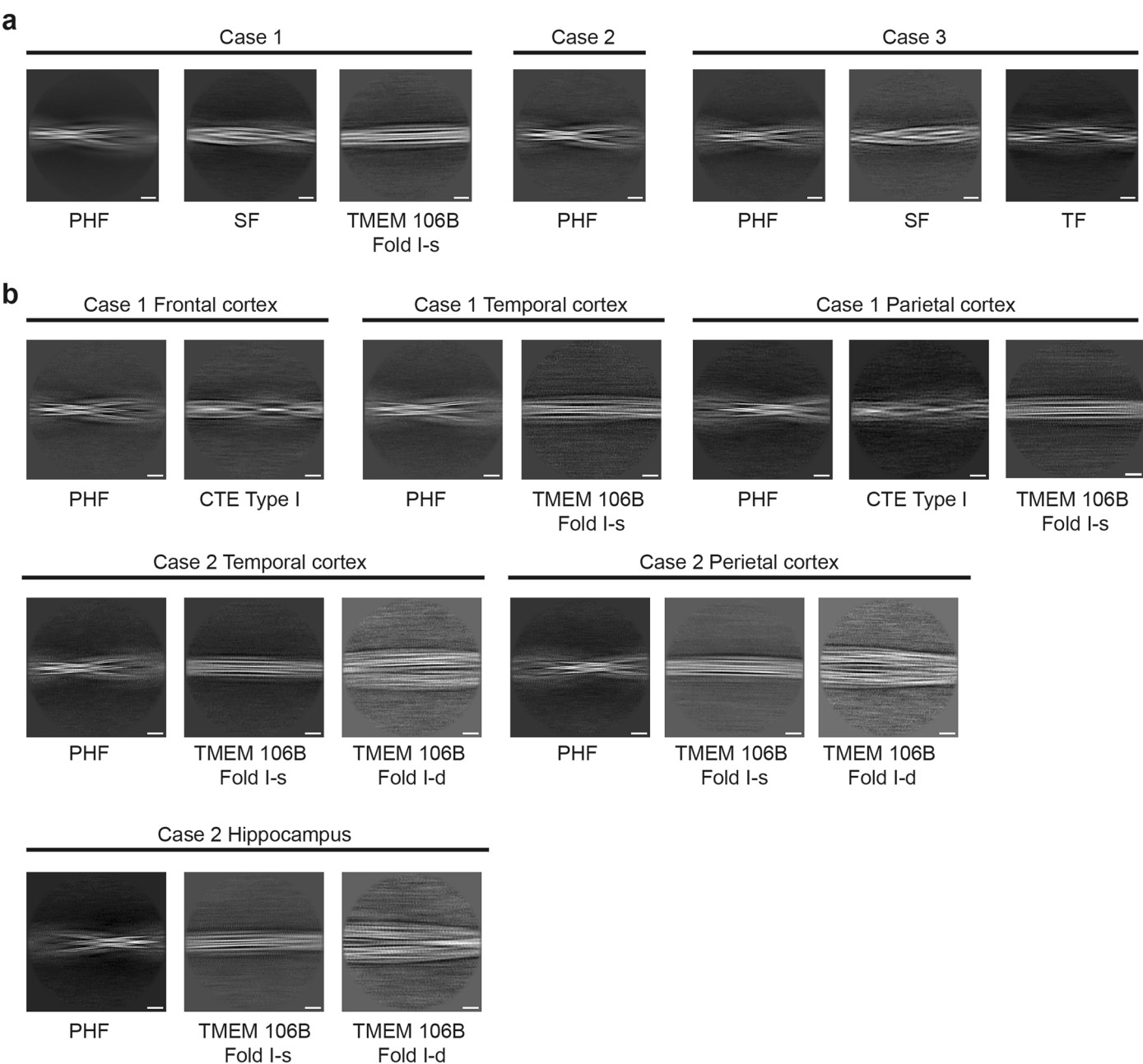

**Extended Data Fig. 8 | Representative 2D class averages. 2D class averages of filaments from three cases with *MAPT* mutation encoding V337M (a) and two cases with *MAPT* mutation encoding R406W (b).** Scale bar, 10 nm.

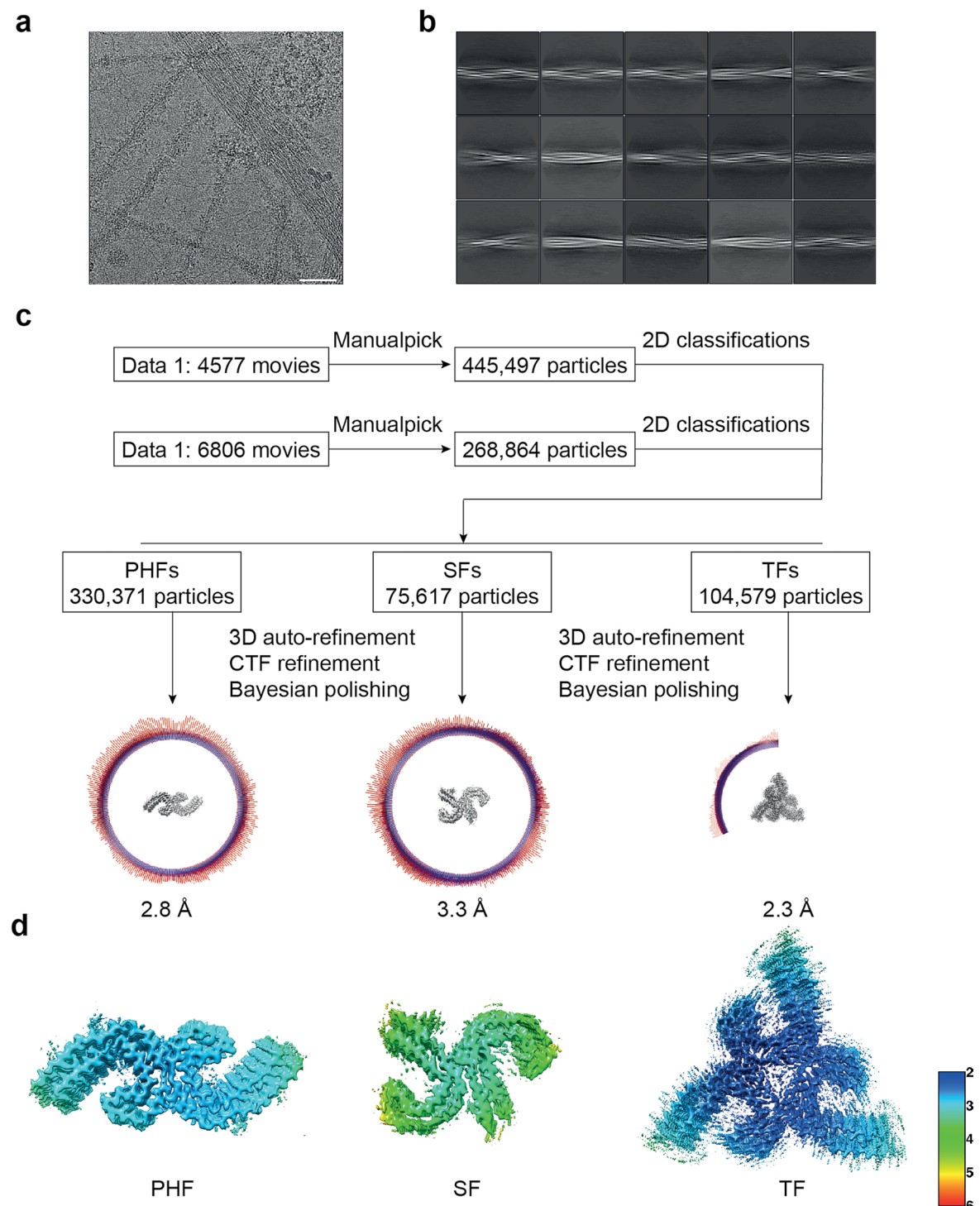

**Extended Data Fig. 9 | Cryo-EM image processing workflow (V337M case3).**
**a**, A representative motion-corrected micrograph; the scale bar, 50 nm.
**b**, Representative 2D classes; the box size is 843 Å. **c**, The overview of the cryo-EM imaging process. Angular distribution histogram of the final reconstruction is represented at the bottom. **d**, Local resolution maps of PHF, SF and TF, estimated using RELION.

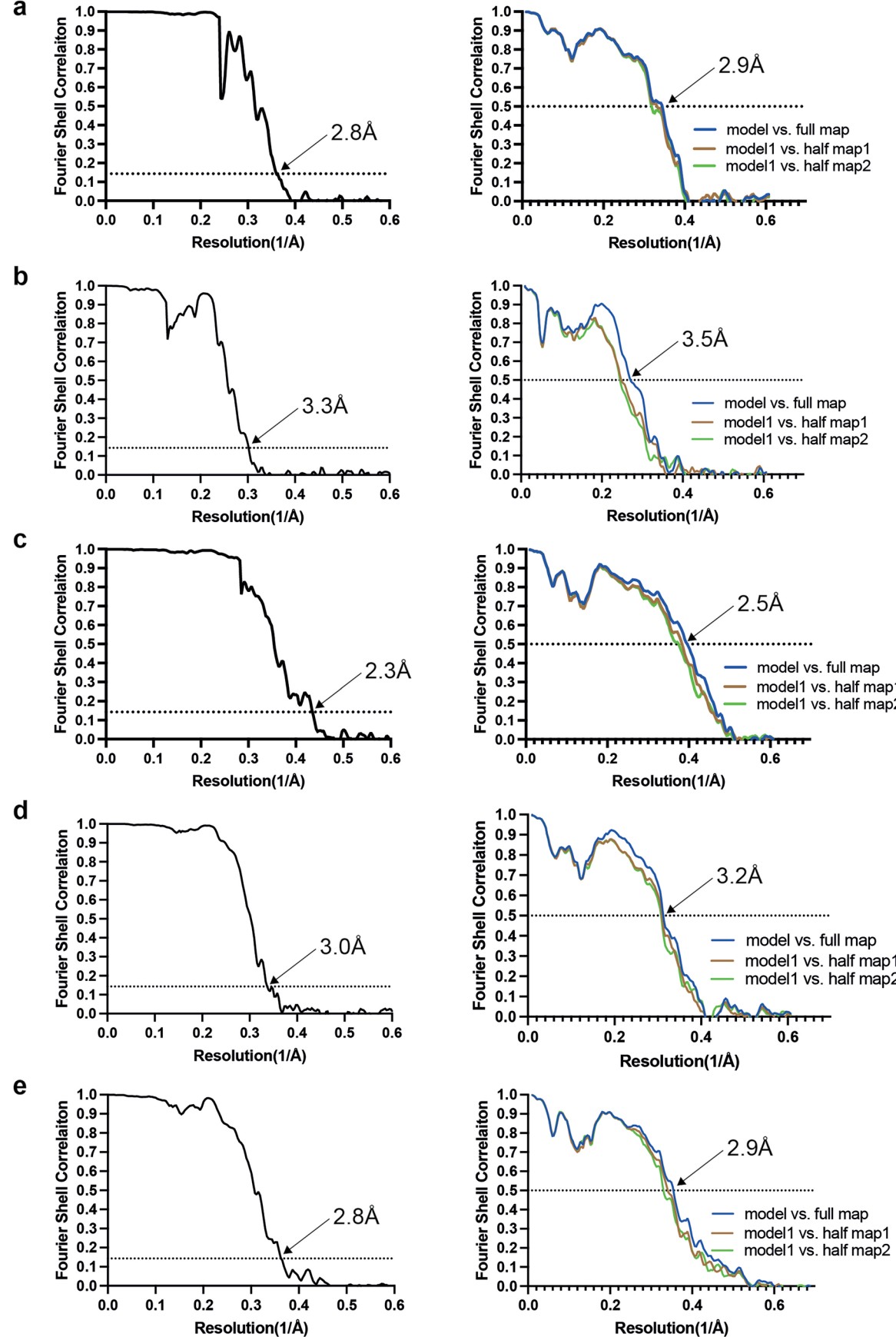

**Extended Data Fig. 10 | Fourier shell correlation (FSC) curves.** FSC curves of cryo-EM maps (left panel) and model to map validation (right panel). **a**, PHF from V337M tau mutant; **b**, SF from V337M tau mutant; **c**, TF from V337M tau mutant; **d**, PHF from R406W tau mutant; **e**, PHF assembled from recombinant V337M tau (297–391).

# Reporting Summary

## Statistics

For all statistical analyses, confirm that the following items are present in the figure legend, table legend, main text, or Methods section.

| n/a | Confirmed | |
|---|---|---|
| ☐ | ☒ | The exact sample size (*n*) for each experimental group/condition, given as a discrete number and unit of measurement |
| ☐ | ☒ | A statement on whether measurements were taken from distinct samples or whether the same sample was measured repeatedly |
| ☒ | ☐ | The statistical test(s) used AND whether they are one- or two-sided *Only common tests should be described solely by name; describe more complex techniques in the Methods section.* |
| ☒ | ☐ | A description of all covariates tested |
| ☒ | ☐ | A description of any assumptions or corrections, such as tests of normality and adjustment for multiple comparisons |
| ☐ | ☒ | A full description of the statistical parameters including central tendency (e.g. means) or other basic estimates (e.g. regression coefficient) AND variation (e.g. standard deviation) or associated estimates of uncertainty (e.g. confidence intervals) |
| ☒ | ☐ | For null hypothesis testing, the test statistic (e.g. $F$, $t$, $r$) with confidence intervals, effect sizes, degrees of freedom and $P$ value noted *Give P values as exact values whenever suitable.* |
| ☒ | ☐ | For Bayesian analysis, information on the choice of priors and Markov chain Monte Carlo settings |
| ☒ | ☐ | For hierarchical and complex designs, identification of the appropriate level for tests and full reporting of outcomes |
| ☒ | ☐ | Estimates of effect sizes (e.g. Cohen's *d*, Pearson's *r*), indicating how they were calculated |

*Our web collection on statistics for biologists contains articles on many of the points above.*

## Software and code

Policy information about availability of computer code

| Data collection | EPU2.3.079 (thermofisher scientific) |
|---|---|
| Data analysis | Relion4, ctffind4.1,chimera1.18, servalcat, refmac5, ISOLDE, MolProbity4.5, COOT 0.9.8.7, ChimeraX 1.6.1 |

For manuscripts utilizing custom algorithms or software that are central to the research but not yet described in published literature, software must be made available to editors and reviewers. We strongly encourage code deposition in a community repository (e.g. GitHub). See the Nature Portfolio guidelines for submitting code & software for further information.

## Data

Policy information about availability of data

All manuscripts must include a data availability statement. This statement should provide the following information, where applicable:

- Accession codes, unique identifiers, or web links for publicly available datasets
- A description of any restrictions on data availability
- For clinical datasets or third party data, please ensure that the statement adheres to our policy

Cryo-EM maps have been deposited in the Electron Microscopy Data Bank (EMDB) with accession numbers: EMD-19846; EMD-19849; EMD-19852; EMD-19854; EMD-19855. Corresponding refined atomic models have been deposited in the Protein Data Bank (PDB) under the following accession numbers: 9EO7; 9EO9; 9EOE; 9EOG; 9EOH.

# Research involving human participants, their data, or biological material

Policy information about studies with <u>human participants or human data</u>. See also policy information about <u>sex, gender (identity/presentation), and sexual orientation</u> and <u>race, ethnicity and racism</u>.

| | |
|---|---|
| Reporting on sex and gender | see method section. 3 cases of Seattle family with mutation V337M, sex: female, famale, male.  1 case US family with mutation R406W, sex:female. 1 case UK family with mutation R406W, sex:male. |
| Reporting on race, ethnicity, or other socially relevant groupings | Not relevant to study. |
| Population characteristics | see method section. 3 cases of Seattle family with mutation V337M, death age 78,63, 58. 1 case US family with mutation R406W, death age 78. 1 case UK family with mutation R406W, death age 66. |
| Recruitment | Selected based on neuropathological examination. |
| Ethics oversight | For the V337M MAPT cases, informed consent for brain donation was obtained from the legal next of kin according to protocols approved by the University of Washington Institutional Review Board that conform to the provisions of the Declaration of Helsinki and preserve donor anonymity. For the R406W MAPT case 1, research protocols for the Indiana Alzheimer's Disease Research Center were approved by the Indiana University Institutional Review Board (protocol:1011003338, initial approval date:04/22/1991, current expiration date:02/05/2025). Brain tissue from R406W MAPT case 2 was donated to the UCL Queen Square Brain Bank with informed consent and the study was approved by the NHS Health Research Authority Ethics Committee, London-Central (REC reference: 23/LO/0044). Genomic DNAs from the V337M and R406W cases were extracted from postmortem brain tissues.  The cryo-EM study was approved by the Cambridgeshire Research Ethics committee (09/HO308/163). |

Note that full information on the approval of the study protocol must also be provided in the manuscript.

# Field-specific reporting

Please select the one below that is the best fit for your research. If you are not sure, read the appropriate sections before making your selection.

☒ Life sciences ☐ Behavioural & social sciences ☐ Ecological, evolutionary & environmental sciences

For a reference copy of the document with all sections, see <u>nature.com/documents/nr-reporting-summary-flat.pdf</u>

# Life sciences study design

All studies must disclose on these points even when the disclosure is negative.

| | |
|---|---|
| Sample size | We used frontal cortex from three previously described cases of Seattle family A with mutation V337M in MAPT. We used temporal and parietal cortex, as well as hippocampus from a female with mutation R406W in MAPT (US family). We used frontal, temporal and parietal cortices from a male with mutation R406W in MAPT (UK family). |
| Data exclusions | No data excluded. |
| Replication | All attempts at replication were successful. |
| Randomization | Because there is no assignment of data points to distinct groups, randomization was not applicable to this study. |
| Blinding | No blinding was performed, as the risk for bias by the experimentalist was deemed irrelevant for this study. |

# Reporting for specific materials, systems and methods

We require information from authors about some types of materials, experimental systems and methods used in many studies. Here, indicate whether each material, system or method listed is relevant to your study. If you are not sure if a list item applies to your research, read the appropriate section before selecting a response.

## Materials & experimental systems

| n/a | Involved in the study |
|---|---|
| ☐ | ☒ Antibodies |
| ☒ | ☐ Eukaryotic cell lines |
| ☒ | ☐ Palaeontology and archaeology |
| ☒ | ☐ Animals and other organisms |
| ☒ | ☐ Clinical data |
| ☒ | ☐ Dual use research of concern |
| ☒ | ☐ Plants |

## Methods

| n/a | Involved in the study |
|---|---|
| ☒ | ☐ ChIP-seq |
| ☒ | ☐ Flow cytometry |
| ☒ | ☐ MRI-based neuroimaging |

## Antibodies

| | |
|---|---|
| Antibodies used | Primary antibodies used are presented in the Methods section. <br> BR134 (Diluted 1:1000) (made by crb Camrbidge Research Biochemicals as request, against human tau C-terminus) <br> AT8 (Diluted 1:1000 or 1:300)(thermofisher scientifc, Catalog # MN1020) <br> RD3 (Diluted 1:3000) (Milipore 05-803, culture supernatant, clone 8E6/C11, Upstate®) <br> RD4 (Diluted 1:100) ((Milipore 05-804,clone 1E1/A6) <br> anti-4R (Diluted 1:400)(Cosmo Bio Catalog No:CAC-TIP-4RT-P01) |
| Validation | BR134 validated against human tau C-terminus in (Goedert et al. 1989 Neuron 3,519-526) <br> AT8 validated against human tau pS202 and pT205 in manufacturer's datasheet (Thermofisher scientific). This Antibody was verified by Cell treatment to ensure that the antibody binds to the antigen stated). <br> RD3 is validated against human 3R tau in manufacturer's datasheet (Millipore) <br> RD4 is validated for use in IH, WB for the detection of Tau (4-repeat isoform RD4). <br> Anti-4R validated against human tau residues 275-291 in (Falcon et al. 2018 Nature 561,137-140) and validated for western blot and IHC(p), this isoform-specific tau antibody is useful for immunohistochemical and biochemical studies of tau species in diverse neurodegenerative diseases. (Cosmo Bio) |

## Plants

| | |
|---|---|
| Seed stocks | Not relevant to study. |
| Novel plant genotypes | Not relevant to study. |
| Authentication | Not relevant to study. |

