## [Peer Review File · Nature Structural & Molecular Biology]

Tau filaments with the Alzheimer fold in cases with MAPT mutations V337M and R406W

Corresponding Author: Professor Michel Goedert

Version 0:

Decision Letter:

10th Jun 2024

Dear Dr. Goedert,

Thank you again for submitting your manuscript "Tau filaments with the Alzheimer fold in cases with MAPT mutations V337M and R406W". We now have comments (below) from the 3 reviewers who evaluated your paper. In light of those reports, we remain interested in your study and would like to see your response to the comments of the referees, in the form of a revised manuscript.

You will see that while reviewers appreciate the results, they raise several concerns which will need to be addressed in a revision. Specifically, we agree with the referees that expanding the analysis of the TFs, as well as more in depth comparisons between the cases will further strengthen the manuscript. We also think that expanding the discussion to address the points brought up by reviewer #3 regarding the uniqueness of filament folds to specific diseases is of interest. Please do attempt comment on the rate of filament assembly in line with reviewer #1 and #3. While we think that the questions brought up by reviewers #1 and #2 about the ability of the observed filaments to seed filaments of specific type are interesting and will benefit the paper, we do not consider them absolutely necessary in the context of the current work to be addressed experimentally, if this proves not feasible.

Please be sure to address/respond to all concerns of the referees in full in a point-by-point response and highlight all changes in the revised manuscript text file. If you have comments that are intended for editors only, please include those in a separate cover letter.

We expect to see your revised manuscript within 6 weeks. If you cannot send it within this time, please contact us to discuss an extension; we would still consider your revision, provided that no similar work has been accepted for publication at NSMB or published elsewhere.

Reporting Summary:

When submitting the revised version of your manuscript, please pay close attention to our [href="https://www.nature.com/nature-portfolio/editorial-policies/image-integrity">Digital Image Integrity Guidelines. and to the following points below:](https://www.nature.com/nature-portfolio/editorial-policies/image-integrity)

If there are additional or modified structures presented in the final revision, please submit the corresponding PDB validation reports. Please also submit any new cryo-EM maps and models which might be obtained during the revision, preferably in a zip folder.

Please note that all key data shown in the main figures as cropped gels or blots should be presented in uncropped form, with molecular weight markers. These data can be aggregated into a single supplementary figure item. While these data can be displayed in a relatively informal style, they must refer back to the relevant figures. These data should be submitted with the final revision, as source data, prior to acceptance, but you may want to start putting it together at this point.

Data availability: this journal strongly supports public availability of data. All data used in accepted papers should be available via a public data repository, or alternatively, as Supplementary Information. If data can only be shared on request, please explain why in your Data Availability Statement, and also in the correspondence with your editor. Please note that for some data types, deposition in a public repository is mandatory - more information on our data deposition policies and available repositories can be found below:

<https://www.nature.com/nature-research/editorial-policies/reporting-standards#availability-of-data>

We require deposition of coordinates (and, in the case of crystal structures, structure factors) into the Protein Data Bank with the designation of immediate release upon publication (HPUB). Electron microscopy-derived density maps and coordinate data must be deposited in EMDb and released upon publication. Deposition and immediate release of NMR chemical shift assignments are highly encouraged. Deposition of deep sequencing and microarray data is mandatory, and the datasets must be released prior to or upon publication. To avoid delays in publication, dataset accession numbers must be supplied with the final accepted manuscript and appropriate release dates must be indicated at the galley proof stage.

Link Redacted

Sincerely,
Kat

Katarzyna Ciazynska, PhD
(she/her)
Associate Editor

Referee expertise:

Referee #1: amyloids, neurobiology

Referee #2: amyloids, structural biology

Referee #3: amyloids, structural biology

Reviewers' Comments:

Reviewer #1:

Remarks to the Author:

In the manuscript entitled "Tau filaments with the Alzheimer fold in cases with MAPT mutations V337M and R406W", Chao et al., determined the Cryo-EM structures of tau filaments from brains of human patients diagnosed with FTD, carrying a tau mutation of V337M or R406W. The authors show that both mutations gave rise to tau filaments with the Alzheimer fold, even though V337M is located inside the ordered core of the filament. They also identified a new assembly of the protofilament into a triple filament in a V337M case. In vitro assembly of a recombinant tau (residues 297-391) indicated that the V337M mutation increases the filament assembly rate, while it has little effect on the conformation of the filaments as demonstrated by the Cryo-EM structures.

It is a very interesting and solid work. It provides an important piece of information to the community that inherited tau single mutants in FTDP-17T can have an Alzheimer fold, at least in the V337M and R406W cases, and certain mutation(s) can accelerate the development of tau filaments, thus the progress of neurodegenerative diseases. It will inspire people in the field and from other disciplines to develop new models and strategies to study the underlying molecular mechanisms of tauopathies and dementia.

Minor comments:

1. Have the authors studied seeded aggregation in cells/neurons using the recombinant tau (297-391) preformed fibril seeds? If so, can these Alzheimer fold tau seeds cause tau inclusions and what the filament structures look like? I am not asking for another structure here. Any comments/discussions from the authors would be helpful to the field.
2. The authors found that mutation V337M increases the assembly rate of recombinant tau (297-391). Could it be because of the presence of ~40% quadruple helical filaments? In relating to the newly identified assembly of triple tau filaments in the V337M case 3, this case developed FTD younger than the other two patients and died earlier. Could it be possible that the appearance of higher order of helical filaments speeds up the tau aggregation and disease severeness?

Reviewer #2:

Remarks to the Author:

In this work, we learn about the structures of tau protein in tau filaments extracted from various brain regions of individuals with V337M or R406W mutations in MAPT (tau) gene, who died from early-onset dementia. All of the studied cases had well-documented history of symptoms, the three V337M cases and one R406W (UK family) case were diagnosed with frontotemporal dementia (FTD), and one R406W (US family) case was diagnosed with Alzheimer's disease (AD). The Alzheimer tau folds dominated in all cases. These consisted of paired helical filaments (PHFs) in all V337M and R406W cases and straight filaments (SFs) in two V337M cases. Interestingly, the authors identified a new ex vivo tau fibril architecture, comprising three tau protofilaments of the Alzheimer fold, but these triple tau filaments (TFs) were found only in one V337M case. One V337M and both R406W cases also presented with TMEM106B filaments. The authors measured the rate of recombinant V337M tau assembly in vitro and found it significantly faster compared to wild-type tau. The structure of the recombinant V337M tau filament was PHF-like, with deviated tau conformation.

It is interesting how variable the three cases of V337M are in their amyloid structure profiles. This tau mutation lies in the amyloid core of tau filaments, but only SFs appear slightly altered around the mutation site, and only in one protofilament. The authors suggest that PHFs can accommodate a relatively long, but flexible methionine side chain in the place of valine, which might explain the apparent lack of perturbation in these brain-derived PHFs, while in vitro polymerised (unseeded) filaments of homogeneous recombinant tau with the same mutation (but lacking post-translational modifications) show altered PHFs. Indeed, the density at the 337 site is typical for M at this resolution only for the in vitro PHF map, and that of the presented ex vivo PHF map looks typical for V, which can also be compared with the neighbouring V density. It is possible, as authors suggest, that M side chain could curl into this confined space, for example by templating imposed by the V variant, since in vivo, the other (wild-type) tau allele is also expressed, as shown by the authors using mass spec, but I think it is also possible that the main or only "sink" for the M variant tau is a single protofilament of the SF type (case 1) or that and TFs (case 3). In this context, I think it would be particularly interesting to closely interrogate the PHFs from V337M

case 2, where these are the only tau filaments detected, however, this case appears under-analysed. Slightly confusingly, in Figure 2b the authors analyse the PHFs of case 1 only (same for SFs in panel d, according to the legend), while the Extended Data Table 1 reports on those of case 3 only, and the maps and model of only the case 3 V337M filaments appear to be made available via database deposition. Also, the authors compare the ex vivo PHF and SF V337M tau fold (case 1) only with that of the equivalent AD fold and not between those from the different V337M cases.

I was also wondering if the authors looked at whether any of the tau assemblies can be found cross-templating other assemblies, forming hybrid fibrils, or whether each filament is strictly of one type. This could be interesting from a mechanistic perspective.

Overall, this work is important and of top quality, but I think the authors do not focus or report enough on the most interesting case in the manuscript which may lead to more insightful conclusions. Perhaps a more in-depth analyses of each mutation in separate manuscripts could be a better way to present all the important work with more elaborate discussion if the current publication format prevents that.

Minor suggestions:

1. I would find it very useful if cross-over distances of all tau assemblies reported could be listed in a table, together with those of AD fibrils.
2. The SF backbone colouring in Figure 2d is hard to distinguish, these shades of green and black/grey might be clearer if the backbones were a bit thicker

Reviewer #3:

Remarks to the Author:

In a workman-like manuscript with interesting implications, the authors report structures of tau amyloid filaments from cases with autosomal dominant tau mutations, V337M and R406W. They find these filaments adopt the AD fold seen previously for wildtype tau in Alzheimer's disease. Mass spectrometry suggests both mutant and wildtype tau proteins are incorporated in the filaments. The V337M mutation is the first to be seen inside the AD core and causes minor structural change – it is accommodated by a 3Å shift along helical axis of beta strand 4 of SF (straight filament) protofilament A and the newly observed triple filament. These filaments are from cases of FTDP-17. Thus FTDP-17 is added to the list of diseases with the AD fold, including PART, familial British and Danish dementias, and prion protein amyloidoses.

Suggestions

No concerns arise about the structural work. However, a limitation of the present manuscript is that it is largely molecular anatomy, and might be better suited to a journal of pathology than one of molecular biology. Here we raise points which, if addressed, might enhance the interest of the work for the community of bioscientists, increasing the fit of the paper for NSMB.

- 1) The authors report several newly observed structural features in tau filaments from FTDP-17. Can they rule out that one or more of these features could be the pathogenic entity FTDP-17?
 - a. The authors state "...cases of FTDP-17 can have the same tau filament fold as cases of AD..." (line 254). Yet, something causes a different disease. The question is what qualifies two structures as being "the same"? To quantitatively evaluate whether V337M tau filament structures are "the same" as the AD fold, the authors can calculate the rmsd of all previously solved AD folds and compare it to the rmsd of V337M tau structures. Importantly, authors should indicate the rmsd for not just the PHF, but the SF and TF as well.
 - b. V337M mutation case 3 contained an unusual ex vivo "triple filament" (TF). In previous studies of AD and other cases with AD fold, TFs have not been reported. The extra density coordinated by K331 of each TF protofilament has not been previously observed (or very prominent) in PHF and SF. Further, the characteristic extra density in PHFs and SFs in contact with K317 and K321 is not present in TF. Are they the same negatively charged cofactor? Can the authors compare extra densities in PHF, SF and TF (by aligning the C-shaped backbones and revealing the extra densities for each polymorph)?
- 2) The rate of filament assembly is faster for recombinant V337M tau compared to wildtype tau, suggesting this mutation could increase aggregation kinetics. Can the authors assess rate of filament assembly for a mixture of V337M and wildtype tau, which more closely resembles the protein population in these cases?
- 3) The interest of the paper might be enhanced if these authoritative authors would add a paragraph to the end of the present Discussion, summarizing the present state of understanding of such questions as the following:
 - a. The question of how mutations V337M and R406W cause FTDP-17 is part of a larger question of whether, and how, amyloid filaments cause neurodegenerative diseases.
 - b. The structures of the tau filaments associated with FTDP-17 seem to be the same, or nearly the same, as those associated with AD. This finding is an additional case in which a single polymorph seems to be associated with more than a single tauopathy, complicating the 1:1 relationship of fold to disease.
 - c. Over 40 tau mutations are associated with FTDP-17. Do the authors expect that cases with tau mutations other than V337M and R406W will also contain tau filaments with AD fold? What are the implications if another tau mutation leads to another filament fold?

Minor points

- 1) In the sentence starting on line 66, we suggest the wording "...suggesting that the physiological function of microtubule binding and pathological assembly are mutually exclusive."
- 2) In the sentence starting on line 254, we suggest the wording "...the same fold can be associated with clinically different conditions."
- 3) In three V337M mutation cases, authors find all different combinations of tau polymorphs PHF, SF and TF. In R406W mutation cases, only PHF fold is observed. Can authors comment on why the appearance of polymorphs is so variable? Is it a matter of sampling rare structural forms?
- 4) The manuscript repeatedly compares FTDP-17 to Alzheimer's disease. The authors may consider adding a few sentences in the Introduction to compare clinical and neuropathological similarities and differences between these diseases.

Signed: David Eisenberg

Version 1:

Decision Letter:

Our ref: NSMB-A49170A

5th Sep 2024

Dear Dr. Goedert,

Thank you for submitting your revised manuscript "Tau filaments with the Alzheimer fold in cases with MAPT mutations V337M and R406W" (NSMB-A49170A). Please accept our apologies for the delay in this decision, which resulted from difficulties in obtaining referee reports. Nevertheless, the paper has now been seen by the original referees and their comments are below. The reviewers find that the paper has improved in revision, and therefore we'll be happy in principle to publish it in Nature Structural & Molecular Biology, pending minor revisions to satisfy the referees' final requests and to comply with our editorial and formatting guidelines. We specifically ask that you incorporate further discussion requested by referee #2.

We are now performing detailed checks on your paper and will send you a checklist detailing our editorial and formatting requirements in about 3 weeks. Please do not upload the final materials and make any revisions until you receive this additional information from us.

To facilitate our work at this stage, it is important that we have a copy of the main text as a word file. If you could please send along a word version of this file as soon as possible, we would greatly appreciate it; please make sure to copy the NSMB account (cc'ed above).

Sincerely,
Kat

Katarzyna Ciazynska, PhD
(she/her)
Associate Editor
Nature Structural & Molecular Biology
<https://orcid.org/0000-0002-9899-2428>

Reviewer #1 (Remarks to the Author):

The authors have addressed all the comments adequately.

Reviewer #2 (Remarks to the Author):

Thank you to the authors for the helpful rebuttal and the revisions. My questions have been sufficiently addressed and I recommend publication.

However, I am still unsure if V337M tau is incorporated into Alzheimer fold PHFs. Indeed, case 2 only shows PHF filaments and MS analysis of the sarkosyl insoluble pellet shows the presence of both the mutant and wild-type tau, but the immunoblot of case 2 sarkosyl insoluble fraction shows high molecular weight material at the top of the blot. Could this be

oligomeric cross-linked tau? If so, could this be composed predominantly of V337M tau? Case 1 convincingly shows V337M tau incorporation into the SF protofilament A and does not show the equivalent high molecular weight signal on the blot. The MS analyses are performed on the whole pellet, would it make sense to analyse the relevant gel pieces instead?

As mentioned earlier, I do not insist on additional experiments, but simply suggest this for the authors' consideration. The manuscript currently implies that the methionine side chain must curl into the valine-like density in the PHFs. If the authors agree that the above could be an alternative explanation, they might consider including it in their manuscript.

The editors have also asked me to comment on the authors' responses to the remarks of referee #3, who has been difficult to reach. I find the responses and the corresponding modifications in the manuscript to be excellent.

Reviewer #3 (Remarks to the Author):

The authors have improved the clarity and significance in their revised ms, which now will be an important addition to the literature.

The authors may consider adding to the supplement the two figures from their rebuttal letter that show, respectively, superposition of PHF and SF structures from different tauopathies, and alignment of PHF, SF, and THF structures which highlight the new extra density coordinated by K331.

Version 2:

Decision Letter:

23rd Jan 2025

Dear Dr. Goedert,

We are now happy to accept your revised paper "Tau filaments with the Alzheimer fold in cases with MAPT mutations V337M and R406W" for publication as an Article in Nature Structural & Molecular Biology.

Your paper will be published online soon after we receive proof corrections and will appear in print in the next available issue. You can find out your date of online publication by contacting the production team shortly after sending your proof corrections.

Sincerely,

Katarzyna Ciazynska, PhD
(she/her)
Senior Editor
Nature Structural & Molecular Biology
<https://orcid.org/0000-0002-9899-2428>

We thank the reviewers for their constructive comments that we address below (in blue).

Reviewer #1:

In the manuscript entitled “Tau filaments with the Alzheimer fold in cases with MAPT mutations V337M and R406W”, Chao et al., determined the Cryo-EM structures of tau filaments from brains of human patients diagnosed with FTD, carrying a tau mutation of V337M or R406W. The authors show that both mutations gave rise to tau filaments with the Alzheimer fold, even though V337M is located inside the ordered core of the filament. They also identified a new assembly of the protofilament into a triple filament in a V337M case. In vitro assembly of a recombinant tau (residues 297-391) indicated that the V337M mutation increases the filament assembly rate, while it has little effect on the conformation of the filaments as demonstrated by the Cryo-EM structures.

It is a very interesting and solid work. It provides an important piece of information to the community that inherited tau single mutants in FTDP-17T can have an Alzheimer fold, at least in the V337M and R406W cases, and certain mutation(s) can accelerate the development of tau filaments, thus the progress of neurodegenerative diseases. It will inspire people in the field and from other disciplines to develop new models and strategies to study the underlying molecular mechanisms of tauopathies and dementia.

Minor comments:

1. Have the authors studied seeded aggregation in cells/neurons using the recombinant tau (297-391) preformed fibril seeds? If so, can these Alzheimer fold tau seeds cause tau inclusions and what the filament structures look like? I am not asking for another structure here. Any comments/discussions from the authors would be helpful to the field.

We are in the process of carrying out experiments using seeds of tau (297-391), with a view to developing biosensor cell lines that develop the Alzheimer tau fold. We expect to write a manuscript describing these experiments later in the year.

2. The authors found that mutation V337M increases the assembly rate of recombinant tau (297-391). Could it be because of the presence of ~40% quadruple helical filaments? In relating to the newly identified assembly of triple tau filaments in the V337M case 3, this case developed FTD younger than the other two patients and died earlier. Could it be possible that the appearance of higher order of helical filaments speeds up the tau aggregation and disease severeness?

This is an interesting hypothesis, which is difficult to test experimentally. We are now mentioning this possibility on pages 7 and 13 of the revised manuscript. On page 7, it says: *‘Interestingly, case 3 with TFs developed FTD at a younger age than cases 1 and 2 without TFs.’* On page 13, it says: *‘It remains to be seen if the formation of QHFs contributed to this effect.’*

Reviewer #2:

In this work, we learn about the structures of tau protein in tau filaments extracted from various brain regions of individuals with V337M or R406W mutations in MAPT (tau) gene, who died from early-onset dementia. All of the studied cases had well-documented history of symptoms, the three V337M cases and one R406W (UK family) case were diagnosed with frontotemporal dementia (FTD), and one R406W (US family) case was diagnosed with Alzheimer's disease (AD). The Alzheimer tau folds dominated in all cases. These consisted of paired helical filaments (PHFs) in all V337M and R406W cases and straight filaments (SFs) in two V337M cases. Interestingly, the authors identified a new *ex vivo* tau fibril architecture, comprising three tau protofilaments of the Alzheimer fold, but these triple tau filaments (TFs) were found only in one V337M case. One V337M and both R406W cases also presented with TMEM106B filaments. The authors measured the rate of recombinant V337M tau assembly *in vitro* and found it significantly faster compared to wild-type tau. The structure of the recombinant V337M tau filament was PHF-like, with deviated tau conformation.

It is interesting how variable the three cases of V337M are in their amyloid structure profiles. This tau mutation lies in the amyloid core of tau filaments, but only SFs appear slightly altered around the mutation site, and only in one protofilament. The authors suggest that PHFs can accommodate a relatively long, but flexible methionine side chain in the place of valine, which might explain the apparent lack of perturbation in these brain-derived PHFs, while *in vitro* polymerised (unseeded) filaments of homogeneous recombinant tau with the same mutation (but lacking post-translational modifications) show altered PHFs. Indeed, the density at the 337 site is typical for M at this resolution only for the *in vitro* PHF map, and that of the presented *ex vivo* PHF map looks typical for V, which can also be compared with the neighbouring V density. It is possible, as authors suggest, that M side chain could curl into this confined space, for example by templating imposed by the V variant, since *in vivo*, the other (wild-type) tau allele is also expressed, as shown by the authors using mass spec, but I think it is also possible that the main or only "sink" for the M variant tau is a single protofilament of the SF type (case 1) or that and TFs (case 3). In this context, I think it would be particularly interesting to closely interrogate the PHFs from V337M case 2, where these are the only tau filaments detected, however, this case appears under-analysed. Slightly confusingly, in Figure 2b the authors analyse the PHFs of case 1 only (same for SFs in panel d, according to the legend), while the Extended Data Table 1 reports on those of case 3 only, and the maps and model of only the case 3 V337M filaments appear to be made available via database deposition. Also, the authors compare the *ex vivo* PHF and SF V337M tau fold (case 1) only with that of the equivalent AD fold and not between those from the different V337M cases.

Case 2 had only PHFs; SFs were not observed. Similarly, when assembling recombinant tau (297-391) with the V337 mutation, we only observed the formation of PHFs. We therefore do not think that SFs formed a sink of tau V337M in the brain. Case 2 was not under-analysed, it merely had only a single tau filament type (PHFs). When we solve identical structures from multiple brains to increase the N-numbers, we typically describe only one representative structure in detail in the Supplementary Tables and produce an atomic model for only that structure. Here we chose the tau filaments from case 3. The structural

variation between tau filaments from the three cases with mutation V337M cases was very small, with rmsds between PHFs being less than 1Å (please see below).

Structural alignment of PHFs from the three V337M cases: case 1 (green), case 2 (orange); case 3 (blue). The rmsd between $C\alpha$ atoms of case 3 and case 1 was 0.295Å; that between $C\alpha$ atoms of case 3 and case 2 was 0.289Å.

The reviewer is correct in pointing out the apparent inconsistency between choosing the PHF from case 1 for Figure 2 and the PHF from case 3 for the Extended Data Table. The reason is that we had mislabelled the legend of Figure 2. Apologies! The corrected legend now reads (Figure 2b):

b, Backbone representation of the overlay of PHFs extracted from the frontal cortex of case 3 with mutation V337M in MAPT (blue) and PHFs extracted from the frontal cortex of an individual with sporadic AD (white, PDB:5O3L). The root mean square deviation (rmsd) between $C\alpha$ atoms of the two structures is 0.767 Å.

We have added the following sentence to the legend of Figure 2f:

The rmsd between $C\alpha$ atoms of TFs and V337M PHFs is 0.653 Å and that between $C\alpha$ atoms of TFs and AD PHFs 0.872 Å.

We have also added the following sentence to the section on 'Model building and refinement' (page 27):

'When multiple maps of the same filament type were resolved, atomic modelling and database submission were only performed for the map with the highest resolution.'

I was also wondering if the authors looked at whether any of the tau assemblies can be found cross-templating other assemblies, forming hybrid fibrils, or whether each filament is strictly of one type. This could be interesting from a mechanistic perspective.

We did not observe hybrid filaments in the extracts from human brains. Studies of the *in vitro* assembly of recombinant tau, following seeding with different types of filaments are ongoing, but they lie outside the scope of the present manuscript.

Overall, this work is important and of top quality, but I think the authors do not focus or report enough on the most interesting case in the manuscript which may lead to more insightful conclusions. Perhaps a more in-depth analyses of each mutation in separate manuscripts could be a better way to present all the important work with more elaborate discussion if the current publication format prevents that.

We disagree with the reviewer who seems to be of the opinion that V337M case 2 was under-analysed and is more interesting than cases 1 and 3. All three individuals with mutation V337M exhibited a majority of PHFs and a minority (in two cases) of SFs. The presence of TMEM106B filaments in case 1 was probably age-related (this individual died aged 78). The findings that mutation V337M in tau gives rise to filaments with the Alzheimer fold and that recombinant V337M tau (297-391) shows an increased rate of filament assembly will be relevant for the design of improved experimental models for Alzheimer's disease.

Minor suggestions:

1. I would find it very useful if cross-over distances of all tau assemblies reported could be listed in a table, together with those of AD fibrils.

We did not measure cross-over distances for individual tau filaments, as our helical averaging procedures do not need this information. Refined helical twist parameters, which can be converted into average cross-over distances, are reported for all representative structures in Extended Data Table 2. They are similar to those observed for Alzheimer's disease.

2. The SF backbone colouring in Figure 2d is hard to distinguish, these shades of green and black/grey might be clearer if the backbones were a bit thicker.

We have now increased the thickness of the backbones by about 50% in Figure 2d.

Reviewer #3:

In a workman-like manuscript with interesting implications, the authors report structures of tau amyloid filaments from cases with autosomal dominant tau mutations, V337M and R406W. They find these filaments adopt the AD fold seen previously for wildtype tau in Alzheimer's disease. Mass spectrometry suggests both mutant and wildtype tau proteins are incorporated in the filaments. The V337M mutation is the first to be seen inside the AD core and causes minor structural change – it is accommodated by a 3Å shift along helical axis of beta strand 4 of SF (straight filament) protofilament A and the newly observed triple filament. These filaments are from cases of FTDP-17. Thus FTDP-17 is added to the list of diseases with the AD fold, including PART, familial British and Danish dementias, and prion protein amyloidoses.

Suggestions

No concerns arise about the structural work. However, a limitation of the present manuscript is that it is largely molecular anatomy and might be better suited to a journal of pathology than one of molecular biology. Here we raise points which, if addressed, might enhance the interest of the work for the community of bioscientists, increasing the fit of the paper for NSMB.

1) The authors report several newly observed structural features in tau filaments from FTDP-17. Can they rule out that one or more of these features could be the pathogenic entity FTDP-17?

a. The authors state "...cases of FTDP-17 can have the same tau filament fold as cases of AD..." (line 254). Yet, something causes a different disease. The question is what qualifies two structures as being "the same"? To quantitatively evaluate whether V337M tau filament structures are "the same" as the AD fold, the authors can calculate the rmsd of all previously solved AD folds and compare it to the rmsd of V337M tau structures. Importantly, authors should indicate the rmsd for not just the PHF, but the SF and TF as well.

We compared the rmsds of the tau filament structures from the V337M cases to those from other conditions with tau filaments (please see below). All rmsd values were smaller than 2 Å, indicating that the same tau fold was present in all conditions.

a

V337M_PHF	rmsd
AD_PHF	0.767Å
PART_PHF	0.292Å
FBD_PHF	0.371Å
FDD_PHF	1.457Å
PCA_PHF	1.433Å
GSS_PHF	1.616Å
PrP-CAA_PHF	1.639Å
DS_PHF	0.490Å

b

V337M_SF	rmsd
AD_SF	0.990Å
PART_SF	0.826Å
FBD_SF	0.478Å
PrP-CAA_SF	0.709Å
DS_SF	0.782Å

Comparison of V337M tau filament structures to those from other conditions.

PHFs (a) and SFs (b) from the frontal cortex of individuals with missense mutation V337M in *MAPT* were compared to those of PHFs and SFs from Alzheimer's disease (AD), primary age-related tauopathy (PART), familial British dementia (FBD), familial Danish dementia (FDD), posterior cortical atrophy (PCA), Gerstmann-Sträussler-Scheinker disease (GSS), cerebral amyloid angiopathy with prion protein deposits (PrP-CAA) and Down's syndrome (DS).

Genetics has established that mutations V337M and R406W cause FTDP-17. They may do so by triggering the assembly of tau into filaments. The present work shows that these filaments have the Alzheimer tau fold; PHFs were found in every V337M and every R406W case. We also show that V337M tau (297-391) assembled into PHFs at a faster rate than tau (297-391). This may be the mechanism by which V337M tau causes FTDP-17. These findings take us beyond mere 'molecular anatomy.'

b. V337M mutation case 3 contained an unusual ex vivo "triple filament" (TF). In previous studies of AD and other cases with AD fold, TFs have not been reported. The extra density coordinated by K331 of each TF protofilament has not been previously observed (or very prominent) in PHF and SF. Further, the characteristic extra density in PHFs and SFs in contact with K317 and K321 is not present in TF. Are they the same negatively charged cofactor? Can the authors compare extra densities in PHF, SF and TF (by aligning the C-shaped backbones and revealing the extra densities for each polymorph)?

Triple tau filaments are unique in that they contain an extra density that is co-ordinated by K331 from each protofilament. The presence of extra

densities appears to be the norm in tau filaments. However, despite our ongoing efforts, we still don't know the molecular identities of those densities. Below we compare the positions of the extra densities in the protofilaments of PHFs, SFs and TFs.

Alignment of protofilaments from PHFs (blue), SFs (green) and TFs (orange).

The positions of the extra densities are indicated. TFs have a unique extra density close to amino acid K331.

2) The rate of filament assembly is faster for recombinant V337M tau compared to wildtype tau, suggesting this mutation could increase aggregation kinetics. Can the authors assess rate of filament assembly for a mixture of V337M and wildtype tau, which more closely resembles the protein population in these cases?

We have added the requested new data to the revised manuscript (Figure 3a).

3) The interest of the paper might be enhanced if these authoritative authors would add a paragraph to the end of the present Discussion, summarizing the present state of understanding of such questions as the following:

a. The question of how mutations V337M and R406W cause FTDP-17 is part of a larger question of whether, and how, amyloid filaments cause neurodegenerative diseases.

Please see above (point 1a).

b. The structures of the tau filaments associated with FTDP-17 seem to be the same, or nearly the same, as those associated with AD. This finding is an additional case in which a single polymorph seems to be associated with more than a single tauopathy, complicating the 1:1 relationship of fold to disease.

We do not believe that there is a 1:1 relationship between fold and disease. Instead, we argue that each disease is characterised by a specific predominating fold, but that a given fold can be found in more than one

disease. Thus, the Alzheimer tau fold is found in Alzheimer's disease, familial British and Danish dementias, cases of Gerstmann-Sträussler-Scheinker disease and PrP-cerebral amyloid angiopathy, primary age-related tauopathy, as well as in cases with missense mutations V337M and R406W in *MAPT*.

c. Over 40 tau mutations are associated with FTDP-17. Do the authors expect that cases with tau mutations other than V337M and R406W will also contain tau filaments with AD fold? What are the implications if another tau mutation leads to another filament fold?

The current total of known *MAPT* mutations in FTDP-17 stands at 65. Of these, mutations V337M and R406W have been shown to be associated with amnesic symptoms like those of Alzheimer's disease. It remains to be seen if other mutations can give rise to the same symptoms and the Alzheimer tau fold. What is already clear is that other *MAPT* mutations lead to other tau folds. The implications are that FTDP-17 is an umbrella term that can now be divided into subgroups.

In response to the reviewer's comments, we have added the following paragraph to the end of the Discussion (page 13):

'In conclusion, mutations in MAPT can give rise to cases of FTDP-17 that resemble sporadic tauopathies, including Pick's disease and AGD. Thus, for mutations with primary effects at the splicing level, the Pick fold forms when filaments are made of wild-type 3R tau (5), whereas the AGD fold forms when filaments are made of wild-type 4R tau (6). Here we show that missense mutations V337M and R406W, which give rise to an amnesic phenotype resembling that of AD, as well as to biochemical changes like those in AD, result in the formation of the Alzheimer tau fold. It follows that cases of FTDP-17 that are caused by mutations in MAPT can be divided into distinct subgroups whose tau filament structures can be defined by cryo-EM.'

Minor points

1) In the sentence starting on line 66, we suggest the wording "...suggesting that the physiological function of microtubule binding and pathological assembly are mutually exclusive."

Done (page 3).

2) In the sentence starting on line 254, we suggest the wording "...the same fold can be associated with clinically different conditions."

Done (page 11)

3) In three V337M mutation cases, authors find all different combinations of tau polymorphs PHF, SF and TF. In R406W mutation cases, only PHF fold is observed. Can authors comment on why the appearance of polymorphs is so variable? Is it a matter of sampling rare structural forms?

Like in Alzheimer's disease, PHFs were the dominant tau filament form in all cases. The proportion of SFs was variable, as in Alzheimer's disease. TFs are new and were only found in one case with mutation V337M. The formation of SFs and TFs may have something to do with genetic background.

4) The manuscript repeatedly compares FTDP-17 to Alzheimer's disease. The authors may consider adding a few sentences in the Introduction to compare clinical and neuropathological similarities and differences between these diseases.

We have added some background information (page 1). The beginning reads now:

Frontotemporal dementia (FTD) and Alzheimer's disease (AD) are the most common forms of early-onset dementia. Unlike AD, FTD begins with behavioural changes, including disinhibition, apathy and impulsiveness, prior to the development of cognitive impairment. Dominantly inherited mutations in MAPT, the microtubule-associated protein tau gene, give rise to cases of FTD and parkinsonism linked to chromosome 17 (FTDP-17). These individuals develop abundant filamentous tau inclusions in brain cells, in the absence of beta-amyloid deposits.

Signed: David Eisenberg

We thank the reviewers for their constructive comments that we address below (in blue).

Reviewer #1:

The authors have addressed all the comments adequately.

Thank you.

Reviewer #2:

Thank you to the authors for the helpful rebuttal and the revisions. My questions have been sufficiently addressed and I recommend publication.

However, I am still unsure if V337M tau is incorporated into Alzheimer fold PHFs. Indeed, case 2 only shows PHF filaments and MS analysis of the sarkosyl insoluble pellet shows the presence of both the mutant and wild-type tau, but the immunoblot of case 2 sarkosyl insoluble fraction shows high molecular weight material at the top of the blot. Could this be oligomeric cross-linked tau? If so, could this be composed predominantly of V337M tau? Case 1 convincingly shows V337M tau incorporation into the SF protofilament A and does not show the equivalent high molecular weight signal on the blot. The MS analyses are performed on the whole pellet, would it make sense to analyse the relevant gel pieces instead?

As mentioned earlier, I do not insist on additional experiments, but simply suggest this for the authors' consideration. The manuscript currently implies that the methionine side chain must curl into the valine-like density in the PHFs. If the authors agree that the above could be an alternative explanation, they might consider including it in their manuscript.

The editors have also asked me to comment on the authors' responses to the remarks of referee #3, who has been difficult to reach. I find the responses and the corresponding modifications in the manuscript to be excellent.

We have added evidence in favour of the presence of M337 in the tau filaments (lines 139-142; line 205). The presence of high-molecular weight tau has been shown time and again in cases with tau filaments, whether sporadic or inherited. This is now mentioned in lines 149-152 and 239-240.

Reviewer #3:

The authors have improved the clarity and significance in their revised ms, which now will be an important addition to the literature.

The authors may consider adding to the supplement the two figures from their rebuttal letter that show, respectively, superposition of PHF and SF structures from different tauopathies, and alignment of PHF, SF, and THF structures which highlight the new extra density coordinated by K331.

Following the suggestion by reviewer 3, we have added a new Extended Data Figure 3. It is discussed in lines 153-156.